# DISCO: Mitigating Bias in Deep Learning with Conditional Distance Correlation

**Emre Kavak** [1 2 3]   **Tom Nuno Wolf** [1 3]   **Christian Wachinger** [1 2 3]

## Abstract

Dataset bias often leads deep learning models to exploit spurious correlations instead of task-relevant signals. We introduce the Standard Anti-Causal Model (SAM), a unifying causal framework that characterizes bias mechanisms and yields a conditional independence criterion for causal stability. Building on this theory, we propose $\text{DISCO}_m$ and sDISCO, efficient and scalable estimators of conditional distance correlation that enable independence regularization in gradient-based models. Across six diverse datasets, our methods consistently outperform or are competitive in existing observed bias mitigation approaches, while requiring fewer hyperparameters and scaling seamlessly to multi-bias scenarios. This work bridges causal theory and practical deep learning, providing both a principled foundation and effective tools for robust prediction. Source Code: https://github.com/yakamoz5/DISCO.

## 1. Introduction

Dataset bias poses a persistent challenge for modern (deep) learning methods (Jones et al., 2023), as it can induce spurious shortcuts that fail to generalize or obscure the task-relevant signal (Geirhos et al., 2020). To systematically understand and address these mechanisms, we operate within an anti-causal prediction setting. In this framework, the target variable of interest, $Y$, is assumed to causally generate the input data $X$ (e.g., an image or tabular data). While our model utilizes $X$ as the input to predict $Y$, the underlying causal flow is $Y \rightarrow X$. Within this setting, several distinct causal pathways can introduce bias.

In Alzheimer's disease prediction, for example, age often

acts as a confounder in a fork structure (see Fig. 1a, where the disease is $Y$ and age is the confounder $B_c$). If left unaddressed, the prediction system may rely predominantly on age-related features rather than identifying actual disease markers or their interaction effects (Zhao et al., 2020). Another widespread phenomenon is collider bias (often referred to as sampling bias in this context), illustrated in Fig. 1b. Collider bias typically arises implicitly through data collection and creates spurious associations between variables that are otherwise causally unrelated. For instance, hospitalization can function as a hidden collider ($B_{col}$) that artificially connects risk factors with diseases ($Y$) without a true underlying mechanism (Griffith et al., 2020).

A third source of prediction instability arises from mediator variables (Fig. 1c). The causal inference literature typically decomposes causal influence into direct and indirect effects (or mediated effects) (Shpitser & VanderWeele, 2011; Pearl, 2012). However, in our anti-causal prediction setup (see Fig. 2), the effect of the target $Y$ on the model output $\hat{Y}$ is inherently mediated by the input features $X$. To avoid terminological ambiguity with standard direct effects, we define the robust, task-relevant signal passing specifically through the $Y \rightarrow X \rightarrow \hat{Y}$ pathway as the *counterfactually stable effect* (ctf-stable). We distinguish this from unwanted shortcuts, which we isolate as spurious indirect effects ($IE$). In many practical applications, mediators carry highly useful and relevant information for the task, especially since they are directed. However, for the purpose of our theoretical analysis, and in specific settings where these intermediate variables represent unstable or irrelevant shortcuts, we treat them as biases that must be controlled (see Section 2.2 for a detailed discussion and exceptions).

To formalize the desired behavior of our model against these biases, we introduce the concept of *causal stability*. Intuitively, a predictor achieves causal stability when its outputs remain robust against counterfactual changes to known bias attributes. By properly regularizing the model, we ensure that it isolates and relies strictly on the robust, causally stable pathways (ctf-stable), effectively stripping away fluctuating spurious associations and unwanted mediated shortcuts.

While such pathway-specific analysis is obvious in classical statistics and causal inference, this structural perspective is largely neglected in the deep learning community. To

[1]Technical University of Munich, Germany [2]Konrad Zuse School of Excellence in Reliable AI, Germany [3]Munich Center for Machine Learning (MCML), Germany. Correspondence to: Emre Kavak <emre.kavak@tum.de>.

*Proceedings of the 43rd International Conference on Machine Learning*, Seoul, South Korea. PMLR 306, 2026. Copyright 2026 by the author(s).

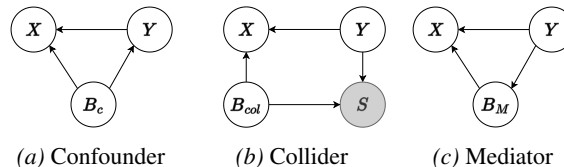

*(a)* Confounder     *(b)* Collider     *(c)* Mediator

*Figure 1.* Canonical causal structures that induce dataset bias. Grey nodes indicate latent conditioning.

formalize how deep learning models can isolate these ctf-stable effects, we introduce the *Standard Anti-Causal Model (SAM)*, a unifying framework that characterizes bias mechanisms in prediction tasks. Specifically, it builds on the *anti-causal prediction setup* (Schölkopf et al., 2012), where the target variable is assumed to generate the input data (Fig. 1). SAM is further inspired by the notations and analyses done by Plecko & Bareinboim (2022) for a path-specific fairness analysis. This graphical framework not only allows us to derive a clear, observable conditional independence criterion for causal stability, but it also serves as a general analytic tool for studying counterfactual bias effects when the full data-generating process is under control.

While prior work has proposed tailored methods to address confounder bias (Neto, 2020; Zhao et al., 2020) or collider bias (Darlow et al., 2020), we deduce from our framework that the specific causal nature of the bias is largely irrelevant: all types can be mitigated uniformly, provided a counterfactually stable (ctf-stable) effect of interest exists.

Specifically, we show that when no unobserved backdoor paths exist between the input $X$ and target $Y$, the conditional independence criterion $\hat{Y} \perp \mathbf{B} \mid Y$, where $\mathbf{B}$ denotes all observed biases and $\hat{Y}$ the model output, mathematically ensures *causal stability*. In this case, all counterfactual indirect and spurious effects ($SE$) vanish, and predictions rely solely on the ctf-stable causal path. When unobserved backdoor paths are present, absolute causal stability can no longer be theoretically guaranteed. However, assuming the input $X$ does not contain valid proxy variables for these hidden confounders (where proximal causal inference methods could otherwise be applied), enforcing this conditional independence criterion remains the theoretical limit and best achievable approximation, as it successfully blocks all biased paths involving the observed $\mathbf{B}$.

Similar independence-based criteria have been used empirically in fairness and bias mitigation (Makar & D'Amour, 2023; Kaur et al., 2022; Puli et al., 2021), and our causal pathway analysis provides a rigorous theoretical foundation for their effectiveness. Related approaches, such as (Quinzan et al., 2022), arrive at partially overlapping conclusions but without leveraging path-specific causal graphical analysis. Compared to Veitch et al. (2021), who prove counterfactual invariance under more restrictive assumptions, our

results generalize to richer anti-causal prediction settings.

Moreover, building on this theoretical foundation, we develop a novel practical approach for enforcing causal stability in black-box models. While conditional distance correlation has been proposed before (Wang et al., 2015), its existing formulations are computationally prohibitive and essentially unusable in modern optimization and deep learning. We resolve this bottleneck by introducing two computational variants, strictly compatible with backpropagation: $DISCO_m$, a computational shortcut but limiting the reference points for the computation to $m$ examples, and a highly efficient variant that first decomposes the original V-statistic of Wang et al. (2015) and performs the computation in a single-shot manner, thus we call it sDISCO (single-shot DISCO). Unlike other shortcut-removal methods, our approach supports arbitrary combinations of target and bias variable types, scales effectively to multivariate settings, and requires significantly fewer hyperparameter choices. Therefore, we successfully enable conditional independence regularization for deep learning models.

In Section 4, we demonstrate across six diverse datasets that $DISCO_m$ and sDISCO consistently perform competitively with or superior to state-of-the-art bias mitigation methods, while having no scaling issues to multi-bias scenarios. Beyond empirical performance, we show how SAM further enables fine-grained pathway analysis of decision processes, highlighting the dual theoretical and practical impact of our work.

### 1.1. Contextualization in CRL and Bias Mitigation

**Causal Representation Learning (CRL).** Our work aligns with the broader goals of Causal Representation Learning (Schölkopf et al., 2021), specifically the objective of learning representations that are robust and generalize in spite of shortcut and bias signals. A full contextualization within CRL can be read in Appendix C.1.

**Bias/Shortcut Mitigation Positioning.** While bias mitigation is an established field, our framework provides a structural foundation that generalizes prior findings. See Appendix C.2 for a full discussion on how our work relates to Makar & D'Amour (2023); Veitch et al. (2021); Puli et al. (2021).

## 2. Causal Invariance

To bridge the gap between our conceptual goals and practical deep learning optimization, we must formally map the flow of task-relevant signals versus spurious biases. In this section, we introduce the Standard Anti-Causal Model (SAM) to characterize these pathways. Using counterfactual analysis, we demonstrate mathematically how total variation in model predictions can be decomposed into stable and un-

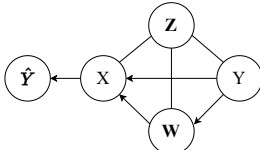

*Figure 2.* SAM graph. $Y$ is the target, $\mathbf{Z}$ are variables on active, non-directed paths between $X$ and $Y$, and $\mathbf{W}$ are mediator variables assumed to be unwanted shortcuts. $\hat{Y}$ denotes the prediction from a fixed prediction model. See appendix Fig. D.1 for more details on permitted relationships.

stable components. From this decomposition, we derive the observational conditional independence criterion required to achieve causal stability. Finally, we critically analyze the underlying assumptions required for this criterion to hold in practice, addressing potential violations such as unobserved confounding.

### 2.1. Standard Anti-Causal Model (SAM) and Causal Stability

Our Standard Anti-Causal Model (SAM) is inspired by the work of Plecko & Bareinboim (2022), and we therefore adopt notation closely aligned with theirs. We define structural causal models (SCMs) as quadruples $(\mathbf{V}, \mathcal{F}, \mathbf{U}, P_{\mathbf{U}})$, where $\mathbf{U}$ are the exogenous variables with distribution $P_{\mathbf{U}}$ defined outside the model. The set $\mathbf{V}$ contains the endogenous random variables, which are generated through deterministic functions of their parents in $\mathbf{V}$ combined with exogenous noise from $\mathbf{U}$. Specifically, $\mathcal{F} = \{f_1, \ldots, f_n\}$ is a set of functions such that $v_i \leftarrow f_i(pa(V_i), U_i)$, where $pa(V_i) \subseteq \mathbf{V}$ are the parents of $V_i$, and $U_i \in \mathbf{U}$ is the exogenous variable associated with $V_i$.

Figure 2 illustrates the SAM considered in this work (we drop the exogenous variables $\mathbf{U}$ for simplicity). It represents a prototypical causal structure that encompasses many relevant scenarios in anti-causal prediction. We partition the variables $\mathbf{V}$ into three groups: the target $Y \in \mathbf{V}$, backdoor variables $\mathbf{Z} \subset \mathbf{V}$ (confounders and variables on open collider paths), and mediator variables $\mathbf{W} \subset \mathbf{V}$. Importantly, in our formulation, any colliders are assumed to be latently conditioned on (e.g., due to implicit sampling or selection bias), which inherently opens backdoor paths. The set $\mathbf{Z}$ explicitly contains the observed nodes that lie on these already-opened paths. For simplicity, independent exogenous noise variables $\mathbf{U}$, latently conditioned colliders, and variables $\mathbf{V}' \subset \mathbf{V}$ that do not open paths between our input $X$ and our target $Y$, are omitted from the figure. While our analysis extends to multidimensional targets, we restrict to a scalar $Y$ for clarity.

For counterfactual and interventional reasoning, we adopt the notation of Pearl (2009) and Plecko & Bareinboim (2022). The expression $P(X_w)$ is equivalent to $P(X \mid$

$do(W = w))$ at the SCM level, while $P(X_w \mid W = w')$ denotes a genuine counterfactual statement. At the unit level, counterfactuals are defined for fixed exogenous realizations $\mathbf{U} = \mathbf{u}$ and denoted $X_w(\mathbf{u})$. These are deterministic functions rather than random variables and are generally non-identifiable. In the following, we assume that SAM holds for our prediction task; later, we discuss violations of these assumptions and their implications.

Throughout our definitions and proofs, we use summation (implicitly assuming the counting measure) for discrete random variables. With appropriate measure-theoretic extensions, these results carry over to continuous variables by replacing sums with integrals.

Our main goal for this section is to establish why the conditional independence criterion presented in the introduction, reading $\hat{Y} \perp \mathbf{B} \mid Y$, is sufficient to mitigate bias, directly deriving this through causal reasoning.

**Measuring Effects of $Y$ on $\hat{Y}$.** A naive way to quantify the influence of $Y$ on $\hat{Y}$ is via the total variation ($TV$) as defined in Plecko & Bareinboim (2022):

$$TV_{y_0, y_1}(\hat{y}) = P(\hat{y} \mid y_1) - P(\hat{y} \mid y_0), \tag{1}$$

where lowercase letters denote realizations $\hat{y}, y_1$, and $y_0$. Note, this is a different TV than the measure-theoretic one. Classical maximum likelihood estimation implicitly maximizes this difference. Assuming binary classification for simplicity, maximizing $P(\hat{Y} = y \mid Y = y)$ corresponds to maximizing the True Positive Rate and minimizing the False Positive Rate, thereby maximizing their difference (TV). However, in the SAM graph, we see that $TV$ captures all types of dependencies, including shortcuts and spurious associations over the pathways $Y$—$\mathbf{Z}$—$\hat{Y}$ and $Y \rightarrow \mathbf{W} \rightarrow \hat{Y}$, rather than only the desired causal influence passing safely through $X$.

**Counterfactual-(Stable, IE, SE)** (Plecko & Bareinboim, 2022). SCMs allow us to specify, and in some cases identify, path-specific effects. While standard causal fairness literature often defines the primary path of interest as a counterfactual direct effect, in our anti-causal setting, the effect of $Y$ on $\hat{Y}$ is strictly mediated by the input $X$. To maintain causal precision and avoid ambiguity, we term this robust target pathway the *counterfactually stable effect* (ctf-stable). Given observations $Y = y$, $\mathbf{Z} = \mathbf{z}$, and $\mathbf{W} = \mathbf{w}$, we let $\mathbf{C} = \{y, \mathbf{w}, \mathbf{z}\}$ denote the conditioning set and define the counterfactually stable (ctf-stable) and indirect (ctf-IE) effects as:

$$ctf\text{-}stable_{y_0, y_1}(\hat{y} \mid \mathbf{C}) = P(\hat{y}_{y_1, w_{y_0}} \mid \mathbf{C}) - P(\hat{y}_{y_0} \mid \mathbf{C}),$$
$$ctf\text{-}IE_{y_1, y_0}(\hat{y} \mid \mathbf{C}) = P(\hat{y}_{y_1, w_{y_0}} \mid \mathbf{C}) - P(\hat{y}_{y_1} \mid \mathbf{C}). \tag{2}$$

Additionally, we define the counterfactual spurious effect (*ctf-SE*) as

$$ctf\text{-}SE_{y_1,y_0}(\hat{y}) = P(\hat{y}_{y_1} \mid y_0) - P(\hat{y}_{y_1} \mid y_1), \quad (3)$$

which captures all non-directed dependencies between $Y$ and $\hat{Y}$.

For notational/mathematical simplicity, without losing any validity of the below statements, we omit the input variable $X$ and treat paths through $X$ as directly reaching $\hat{Y}$.

**Proposition 2.1** (TV-Decomposition). *The total variation can be decomposed into stable, indirect, and spurious components as*

$$TV_{y_0,y_1}(\hat{y}) = \sum_{\mathbf{w},\mathbf{z}} P(\mathbf{w}, \mathbf{z} \mid y) \big[ \text{ctf-stable}_{y_0,y_1}(\hat{y} \mid \mathbf{C}) $$
$$- \text{ctf-IE}_{y_1,y_0}(\hat{y} \mid \mathbf{C}) \big] - \text{ctf-SE}_{y_1,y_0}(\hat{y}),$$
$$(4)$$

*where* $\mathbf{C} = \{y, \mathbf{w}, \mathbf{z}\}$ *denotes the conditioning set. The proof is given in Appendix E.1.*

We assume that the only reliable relationship under distribution, covariate, or domain shifts is the counterfactually stable effect $ctf\text{-}stable_{y_0,y_1}$ (i.e., $Y \to X \to \hat{Y}$). We now identify an observational criterion that ensures both $ctf\text{-}IE_{y_1,y_0}$ and $ctf\text{-}SE_{y_1,y_0}$ vanish.

**Definition 2.2** (Causal Stability). *A predictor is* causally stable *if for all relevant variables and realizations it satisfies*

$$ctf\text{-}IE_{y_1,y_0}(\hat{y} \mid y, \mathbf{w}, \mathbf{z}) = 0 \quad \text{and} \quad ctf\text{-}SE_{y_1,y_0}(\hat{y}) = 0.$$
$$(5)$$

**Theorem 2.3** (Cond. Independence $\Rightarrow$ Causal Stability). *If a model's predictions satisfy* $\hat{Y} \perp \mathbf{W}, \mathbf{Z} \mid Y$ *under SAM, then the model is* causally stable. *The proof is provided in Appendix E.2.*

Note that causal stability alone does not guarantee predictive usefulness. For example, a constant predictor that outputs the same $\hat{Y}$ irrespective of the input trivially satisfies causal stability, but carries no meaningful predictive signal. The following result shows how to obtain both stability and a strong predictive signal.

**Theorem 2.4** (Maximizing $ctf\text{-}stable_{y_0,y_1}$). *A maximum likelihood (MLE) predictor that satisfies* $\hat{Y} \perp \mathbf{W}, \mathbf{Z} \mid Y$ *also maximizes the counterfactually stable effect* $ctf\text{-}stable_{y_0,y_1}$. *The proof is provided in Appendix E.2.*

**Corollary 2.5** (Indirect vs. Spurious Effects). *There is no need to distinguish between mediated and spurious bias effects when analyzing causal stability. By defining the set of all bias variables as* $\mathbf{B} = \mathbf{W} \cup \mathbf{Z}$, *the conditional independence criterion reduces to* $\hat{Y} \perp \mathbf{B} \mid Y$.

Thus, to obtain a stable predictor that relies solely on the stable pathway from $Y$ to $X$, we can formulate the following constrained optimization problem:

$$\min_{\theta} \quad \mathbb{E}_{(X,Y)} \left[ L\left(Y, g_\theta(X)\right) \right]$$
$$\text{s.t.} \quad \hat{Y} \perp \mathbf{B} \mid Y,$$
$$(6)$$

where $\hat{Y} = g_\theta(X)$, $g$ is a learning model parametrized by $\theta$, and $L$ is a standard loss function (e.g., mean squared error, cross-entropy) corresponding to an MLE under appropriate noise assumptions.

Equation 6 presents an idealized theoretical objective. However, successfully translating this mathematical guarantee into practical deep learning relies on specific properties of the data-generating process. Before introducing our computational estimators for this objective in Section 3.1, we must first examine the core assumptions underlying SAM and discuss the practical implications when these conditions are violated.

### 2.2. Analyzing Assumptions of SAM

**Sufficient Variability (Positivity).** A fundamental requirement for our method, and indeed for any observational bias mitigation approach, is the assumption of sufficient variability or *overlap*. Analogous to the positivity assumption in causal inference, we assume that for any target realization $y$, the data distribution has support over the bias attributes $b$, i.e., $P(B = b \mid Y = y) > 0$. For example, to disentangle the spurious correlation between 'waterbird' and 'water background', the dataset must contain, however rarely, counter-examples such as waterbirds in land environments. Consequently, if $B$ is a deterministic function of $Y$ (zero overlap), the confounding is non-identifiable from observational data alone and debiasing the predictions is generally impossible without further guidance. This limitation applies to all bias mitigation methods.

**Selective Mediation.** While our framework treats mediators as sources of bias by default, practical applications often involve mediators that carry robust, task-relevant causal signals. In such cases, we can distinguish between unstable mediators and a subset of useful mediators $\mathbf{W}_{stable} \subset \mathbf{W}$. We then redefine the regularization bias set as $\mathbf{B}' = (\mathbf{W} \setminus \mathbf{W}_{stable}) \cup \mathbf{Z}$. Our theoretical results transfer directly to this setting: the model is regularized to be independent of $\mathbf{Z}$ and unstable mediators, while remaining free to utilize information from $\mathbf{W}_{stable}$.

**Unobserved Confounding & Proxies.** Finally, an important violation of our assumptions arises from unobserved pathways of information. Since confounders and other spurious effects are modeled by $\mathbf{Z} \subseteq \mathbf{B}$, the absence of some confounders means we work with an incomplete set of observations. The true set of spurious attributes is then $\mathbf{Z}_{true} = \mathbf{Z} \cup \mathbf{Z}'$, where $\mathbf{Z}'$ represents unobserved confounders. In this case, even an MLE predictor constrained

by $\hat{Y} \perp \mathbf{B} \mid Y$ remains biased, as information may still leak through unobserved paths involving $\mathbf{Z}'$. However, strictly assuming that the input $X$ does not contain valid proxy variables for these unobserved confounders, the criterion $\hat{Y} \perp \mathbf{B} \mid Y$ remains the best achievable approximation of causal stability, as it guarantees all observed paths involving $\mathbf{B}$ remain blocked. If valid proxies are available, proximal causal inference methods (Miao et al., 2018; Tchetgen et al., 2020) serve as a complementary approach to recover identifiability.

# 3. Distance Correlation for Conditional Independence

In this section, our main goal is to establish a practical method that enables black-box predictors, such as deep neural networks, to attain causally stable solutions. Intuitively, causal stability requires a model's predictions to be invariant to spurious pathways and rely solely on the true task signal. To enforce this, we must optimize towards the conditional independence criterion defined in Eq. 6: the model's predictions must be independent of the bias attributes, conditioned on the true target.

Traditional linear metrics like conditional covariance are insufficient here, as neural networks learn highly non-linear representations. Distance correlation (Székely et al., 2007) provides a powerful alternative, capable of quantifying non-linear statistical dependence between arbitrary random vectors. Subsequent work (Póczos & Schneider, 2012; Wang et al., 2015; Pan et al., 2017) extended this concept to the conditional setting. However, applying exact conditional distance correlation to deep learning presents a severe computational bottleneck: naively computing the local dependencies for a batch of size $n$ requires allocating an $\mathcal{O}(n^3)$ tensor, which quickly exhausts GPU memory.

We address this algorithmic gap by using two differentiable conditional distance correlation computations: $\text{DISCO}_m$ and sDISCO. While $\text{DISCO}_m$ samples local neighborhoods to reduce memory, our primary contribution, *single-shot* sDISCO, introduces an exact algebraic factorization that computes the rigorous conditional distance correlation across all points simultaneously while strictly confining memory to $\mathcal{O}(n^2)$. Finally, we demonstrate how these estimators can be seamlessly integrated into standard training pipelines to penalize shortcut learning.

## 3.1. Background

To establish the theoretical properties of our conditional independence estimators, we build upon the work of Lyons (2013), who worked on the unconditional case.

Let $(\mathcal{X}, d_\mathcal{X})$, $(\mathcal{Y}, d_\mathcal{Y})$, and $(\mathcal{Z}, d_\mathcal{Z})$ be metric spaces. Let $(X, Y, Z)$ be random elements defined on a common prob-

ability space, taking values in $\mathcal{X} \times \mathcal{Y} \times \mathcal{Z}$, with joint distribution $P$. Denote by $P_{(X,Y)|Z}$ the regular conditional distribution of $(X, Y)$ given $Z$. For each $z \in \mathcal{Z}$, let $\theta_z := P_{(X,Y)|Z=z}$ be a Borel probability measure on $\mathcal{X} \times \mathcal{Y}$.

**Definition 3.1** (Finite first moments). We say that $P_{(X,Y)|Z=z}$ has finite first moments if

$$\int d_\mathcal{X}(x, o_\mathcal{X}) + d_\mathcal{Y}(y, o_\mathcal{Y}) \, d\theta_z(x, y) < \infty \quad (7)$$

for some (and hence all, by the triangle inequality) base points $o_\mathcal{X} \in \mathcal{X}$ and $o_\mathcal{Y} \in \mathcal{Y}$.

**Definition 3.2** (Conditional distance covariance). Let $P_Z$ denote the distribution of $Z$. For $z \in \mathcal{Z}$, let $\mu_z := P_{X|Z=z}$ and $\nu_z := P_{Y|Z=z}$ be the conditional marginals. For $\theta_z$, define the distance-centered kernels

$$d_{\mu_z}(x, x') := d_\mathcal{X}(x, x') - a_{\mu_z}(x) - a_{\mu_z}(x') + D(\mu_z), \quad (8)$$

where $a_{\mu_z}(x) := \int d_\mathcal{X}(x, x') \, d\mu_z(x')$ and $D(\mu_z) := \int \int d_\mathcal{X}(x, x') \, d\mu_z(x) d\mu_z(x')$. The definition of $d_{\nu_z}$ is analogous.

The conditional distance covariance is then given by

$$\text{dCov}^2(X, Y \mid Z) := \mathbb{E}_Z \left[ \text{dCov}^2(X, Y \mid Z = z) \right], \quad (9)$$

with

$$\begin{aligned} \text{dCov}^2(X, Y \mid Z = z) := \int & d_{\mu_z}(x, x') \, d_{\nu_z}(y, y') \\ & \times d\theta_z(x, y) \, d\theta_z(x', y'). \end{aligned} \quad (10)$$

**Theorem 3.3** (Conditional independence and strong negative type). *Assume $\mu_z$, $\nu_z$, and $\theta_z$ have finite first moments. Further suppose that the metric spaces $\mathcal{X}$ and $\mathcal{Y}$ are of strong negative type[1]. Then*

$$\begin{aligned} &\text{dCov}^2(X, Y \mid Z) = 0 \\ &\iff P_{(X,Y)|Z=z} = P_{X|Z=z} \otimes P_{Y|Z=z} \quad (11) \\ &\quad \text{for } P_Z\text{-almost every } z. \end{aligned}$$

*That is, $X$ and $Y$ are conditionally independent given $Z$ almost surely. A proof, together with full definitions and necessary lemmas, is provided in Appendix F.*

Since distance covariance is unbounded, we instead use its correlation analogue, which lies in $[0, 1]$ and is more optimization-friendly.

**Definition 3.4** (Conditional distance correlation). The *conditional distance correlation* between $X$ and $Y$ given $Z$ is

---

[1]Strong negative type ensures that distance-based measures such as distance covariance uniquely identify independence; see Appendix F.

defined as

$$\mathrm{dCor}^2(X, Y \mid Z) := \frac{\mathrm{dCov}^2(X, Y \mid Z)}{\sqrt{\mathrm{dVar}^2(X \mid Z)\, \mathrm{dVar}^2(Y \mid Z)}}. \tag{12}$$

with the convention $0/0 := 0$.

## 3.2. Sample Estimation

Let $\{(X_i, Y_i, Z_i)\}_{i=1}^n$ be an i.i.d. sample in Euclidean spaces. We use the Euclidean distance $d(x, x') = \|x - x'\|$, which is of strong negative type (Sejdinovic et al., 2013).

To construct our estimators, we fix a positive-definite kernel $K_h : \mathcal{Z} \times \mathcal{Z} \to \mathbb{R}_{\geq 0}$ with bandwidth $h$ (e.g., an RBF kernel). We define the row-normalized weight matrix $W = [w_{ij}]$ such that $w_{ij} = K_h(Z_i, Z_j) / \sum_k K_h(Z_i, Z_k)$, which acts as an empirical estimator for the conditional probability density. We also compute the pairwise distance matrices $A = [a_{ij}]$ and $B = [b_{ij}]$ where $a_{ij} = d_{\mathcal{X}}(X_i, X_j)$ and $b_{ij} = d_{\mathcal{Y}}(Y_i, Y_j)$.

**Method 1: DISCO$_m$.** The standard approach to compute the V-statistic of conditional distance covariance evaluates the metric locally (Wang et al., 2015). For a specific reference point $Z_i$, we use the corresponding weight vector $w^{(i)}$ (the $i$-th row of $W$) to construct the locally centered matrices:

$$A_{k\ell}^{(i)} = a_{k\ell} - \sum_k^n w_k^{(i)} a_{k\ell} - \sum_\ell^n w_\ell^{(i)} a_{k\ell} + \sum_{k,\ell}^n w_k^{(i)} w_\ell^{(i)} a_{k\ell}, \tag{13}$$

and define $B_{k\ell}^{(i)}$ analogously. The local conditional squared distance covariance $\mathcal{V}_{XY}^{(i)}$ at $Z_i$ is evaluated as the weighted inner product of these centered matrices:

$$\mathcal{V}_{XY}^{(i)} = \sum_{k=1}^n \sum_{\ell=1}^n w_k^{(i)} w_\ell^{(i)} A_{k\ell}^{(i)} B_{k\ell}^{(i)}. \tag{14}$$

To manage memory, the DISCO$_m$ estimator averages these local correlations over a randomly sampled subset of $m \leq n$ reference points:

$$\mathrm{DISCO}_m(X, Y \mid Z) = \frac{1}{m} \sum_{i=1}^m \frac{\mathcal{V}_{XY}^{(i)}}{\sqrt{\mathcal{V}_{XX}^{(i)} \mathcal{V}_{YY}^{(i)}}}. \tag{15}$$

**Proposition 3.5.** *Building on standard kernel regression assumptions, the DISCO$_m$ estimator provides a consistent estimation of* $\mathrm{dCor}^2(X, Y \mid Z)$. *See Appendix F.3 that shows consistency of this estimator.*

**Method 2: sDISCO (Single-Shot DISCO).** While DISCO$_m$ is effective, naively evaluating the exact global correlation ($m = n$) requires broadcasting the matrices to

shape $(n, n, n)$. This $\mathcal{O}(n^3)$ memory footprint is prohibitive for standard deep learning batch sizes. To overcome this, we propose sDISCO, which factorizes the mathematical operations to achieve the exact global correlation in a single step while strictly confining memory to $\mathcal{O}(n^2)$.

We exploit that the weighted marginal sums of locally centered matrices $A^{(i)}$ and $B^{(i)}$ are exactly zero; when expanding their inner product, cross-terms involving isolated marginal means naturally vanish. While Wang et al. (2015) used this expansion to theoretically decompose the V-statistic into three components ($D_1, D_2, D_3$) to prove estimator equivalence, we leverage it computationally. The problem simplifies to extracting the diagonals of implicit quadratic forms, which can be executed simultaneously for all $n$ points using the Hadamard product ($\circ$) and dense matrix multiplications.

First, we compute local row means: $M^X = WA$ and $M^Y = WB$. Next, we extract local grand means: $g^X = (W \circ M^X)\mathbf{1}$ and $g^Y = (W \circ M^Y)\mathbf{1}$, where $\mathbf{1}$ is a vector of ones. The localized covariances for all $n$ reference points evaluate exactly to the sum of three terms (mapping directly to $D_1$, $D_2$, and $D_3$):

$$T_1 = (W \circ (W(A \circ B)))\mathbf{1}, \tag{16}$$

$$T_2 = g^X \circ g^Y, \tag{17}$$

$$T_3 = (W \circ M^X \circ M^Y)\mathbf{1}. \tag{18}$$

The vector containing all exact local squared covariances is simply $\mathcal{V}_{XY} = T_1 + T_2 - 2T_3$. Applying this symmetrically to obtain variances $\mathcal{V}_{XX}$ and $\mathcal{V}_{YY}$, the global sDISCO estimator is the average of the exact local correlations.

**Proposition 3.6.** *Building on the properties of V-statistics with random kernels, sDISCO calculates the exact sample conditional distance correlation* $\mathrm{dCor}^2(X, Y \mid Z)$ *uniformly in* $\mathcal{O}(n^2)$ *memory. See Appendix G for the proof.*

## 3.3. Practical Application for Causal Stability

Because solving the constrained optimization in Eq. 6 directly is infeasible, we adopt a regularization approach. Substituting the variables from SAM, namely the true target $Y$, the observed bias attributes $\mathbf{B}$, and the model's continuous predictions $\hat{Y} = g_\theta(X)$, we jointly minimize the prediction loss and the conditional dependence penalty:

$$\min_\theta \sum L\left(Y, \hat{Y}\right) + \lambda \, \mathrm{sDISCO}(\hat{Y}, \mathbf{B} \mid Y). \tag{19}$$

This framework aligns seamlessly with black-box, gradient-based training. Importantly, during the forward pass and inference, the model $g_\theta$ generates its prediction $\hat{Y}$ using only the input $X$; it does not observe the bias $\mathbf{B}$. The bias variables are utilized strictly during training in the backward pass to compute the DISCO penalty, forcing the network

to unlearn representations that leak bias information. Both estimators require a kernel bandwidth parameter $\sigma_Y$ over the conditioning variable $Y$, and a hyperparameter $\lambda$ to control regularization strength. For $\text{DISCO}_m$, we fix $m$ to be 20 percent of the batch size, while sDISCO natively processes the full batch.

We show how sDISCO is non-trivial when it comes to compute times and memory requirements in deep learning settings, see Section H.5.

# 4. Experiments

In this section, we evaluate the efficacy of $\text{DISCO}_m$ and sDISCO for bias mitigation. We design experiments on six datasets (vision and NLP) that vary in realism, bias sources, and the non-linear relationships between target and bias attributes. This creates a diverse and challenging testbed for assessing causal stability. We benchmark against seven representative baselines, detailed below, and additionally leverage SAM for pathway-specific counterfactual analysis in fully controlled simulation settings. This allows us to assess not only empirical performance but also the causal properties of our methods in the presence of known and unknown biases.

## 4.1. Datasets and Experimental Setup

Figure 3 illustrates the causal structures underlying our datasets. Across all datasets, except MNLI, we follow the standard evaluation protocol in domain generalization and bias mitigation (Sagawa et al., 2019; Arjovsky et al., 2019; Ganin et al., 2016): we train on a biased dataset, select models on an unbiased validation set, and report final results on an unseen unbiased test set. For these datasets, we additionally add label noise to make the task harder and to make models to latch on spurious correlations more easily. The full details for our MNLI dataset can be found in the Appendix H.1.6 as this dataset deviates from the rest in important aspects.

**dSprites.** Based on Matthey et al. (2017), we construct images of geometric shapes (Sh) with different orientation (O), scale (Sc), y-position (Y), and x-position (X). The regression target, Y, nonlinearly determines the object's y-position while X acts as a confounder, see Section H.1.1.

**Blob dataset.** Inspired by Adeli et al. (2021), we generate synthetic images with two Gaussian blobs. One blob is causally related to the regression target, causal intensity (CI), while the other acts as a spurious mediator, bias intensity (BI). All details can be read in Section H.1.2.

**YaleB.** From the extended YaleB dataset (Georghiades et al., 2001; Yale, 2001), we predict face pose (collapsed

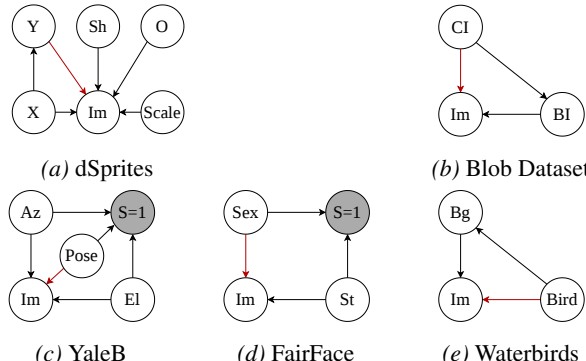

*Figure 3.* Causal graphs used in our experiments across different datasets. Blue path means causally task-relevant, going from target to our input (image).

| | Bias Type | | | Target Type | | | Attr. | |
|---|---|---|---|---|---|---|---|---|
| Method | B | C | ℝ | B | C | ℝ | 1 | M |
| GDRO, Fishr, IRM | ✓ | ✓ | ✗ | ✓ | ✓ | ✗ | ✓ | ✓ |
| C-MMD | ✓ | ~ | ✗ | ✓ | ~ | ✗ | ✓ | ~ |
| Adversarial | ✓ | ✓ | ✓ | ✓ | ✓ | ✓ | ✓ | ~ |
| DISCO, CIRCE, HSCIC | ✓ | ✓ | ✓ | ✓ | ✓ | ✓ | ✓ | ✓ |

*Table 1.* Capabilities summary. **B**: Binary, **C**: Categorical, ℝ: Continuous. **1**: Single-attribute, **M**: Multi-attribute. ✓: Supported, ~: Limited.

into three categories: frontal, slightly left, maximally left). Azimuth (Az) and elevation (El) of the light source serve as continuous bias variables, see Section H.1.3.

**FairFace.** We use FairFace (Karkkainen & Joo, 2021) to predict sex as the target. Skin tone (light vs. dark) acts as a bias through selection bias. See Section H.1.4 for all details, and 6 for ethical considerations.

**Waterbirds.** We adopt the Waterbirds dataset (Sagawa et al., 2019), constructed by overlaying birds (Welinder et al., 2010; Wah et al., 2011) on land or water backgrounds (Bg) (Zhou et al., 2017), see Section H.1.5.

**MNLI.** Finally, we also include the MNLI dataset (Williams et al., 2018). In this dataset, the task is to predict entailment of a sentence given another one. The bias in this task are negation keywords, that are highly correlated with entailment. We closely follow Sagawa et al. (2019) in designing our experiments. Full details for this setup are given in the Section H.1.6.

## 4.2. Bias Mitigation Baselines

To have a rigorous comparison, we benchmark against seven baselines: GDRO (Sagawa et al., 2019), adversarial learning (Ganin et al., 2016; Wang et al., 2019; Adeli et al., 2021), and three recent dependence-penalization methods: HSCIC

Table 3. Counterfactual sensitivity measures.

| Model | $S_X \downarrow$ | $S_Y \downarrow$ | $Acc_{ctf} \uparrow$ |
|---|---|---|---|
| sDISCO$_{X,Y}$ | **0.066** | **0.067** | **0.98** |
| sDISCO$_X$ | 0.117 | 0.154 | 0.83 |
| ResNet | 0.302 | 0.275 | 0.65 |

Figure 4. Multi Bias Scenario dSprites.

(Quinzan et al., 2022), CIRCE (Pogodin et al., 2022), and c-MMD (Kaur et al., 2022; Makar & D'Amour, 2023; Veitch et al., 2021). We further adapt two models from the domain generalization field, namely Fishr (Rame et al., 2022) and IRM (Arjovsky et al., 2019), to operate on groups, exactly as GDRO does. All methods share the same backbone (ResNet-18, pretrained on ImageNet (He et al., 2016; Deng et al., 2009) for real images, untrained for synthetic datasets). For the Blob dataset, we use a smaller ResNet adapted to low resolution. For MNLI, we use the pretrained TinyBERT from Jiao et al. (2020) as the backbone.

To ensure a fair comparison, we assembled the hyperparameter grid ranges for each method from existing work. We additionally gave most methods a larger hyperparameter grid than to our own ones. And some methods, especially IRM and Fishr, needed some adjustment to fit our non-environment bias mitigation setup with groups. We describe all details in the appendix H.2 and the hyperparameter grids can be found in Tab. 5.

### 4.3. Results

The first obvious advantage of our DISCO variants are their applicability for regressions and classification tasks, while being able to handle multi-variable, multi-type bias variable scenarios efficiently (see Tab. 1).

Tab. 2 shows that the base backbone networks without de-biasing suffer from massive performance drops on the balanced test set. The "No Bias" baselines show what the backbones could theoretically achieve when the training data would have been unbiased in the first place. We further see that DISCO$_m$ and sDISCO consistently outperform most baselines, and are competitive with or superior to GDRO and CIRCE. DISCO$_m$ achieves the best results on dSprites and FairFace, while sDISCO ranks second. CIRCE is strong on Waterbirds, which might be explained by the higher hyperparameter search budget we assigned to it. HSCIC and adversarial training generally lag behind. For MNLI, sDISCO performs best.

To ensure a rigorous and fair comparison, we established the

hyperparameter search spaces for all baseline methods based on prior literature. Notably, we allocated broader search grids to most baselines compared to those evaluated for our DISCO variants. Furthermore, certain methods, specifically IRM and Fishr, required adaptations to accommodate our group-based, rather than environment-based, bias mitigation framework. Comprehensive details regarding these modifications are provided in Appendix H.2, and the complete hyperparameter grids are summarized in Table 5.

### 4.4. Path Analysis and Unobserved Bias

While classical bias mitigation experiment results are reported on an unbiased test set, one can also perform pathway analysis if we have access to the data-generating process. These pathway-specific effects allow us more in-depth insights into the causal stability of the tested methods. To illustrate the analytic power of SAM, we conduct pathway-specific counterfactual analysis in a controlled setting on the dSprites dataset, see (Figure 4), as we have full control over the dataset, including the generation of true counterfactuals. We compare a naive ResNet to two variants of sDISCO: one aware of both bias variables (sDISCO$_{XY}$) and one aware of only one bias (sDISCO$_X$), thus the latter model will be simulated in an unobserved confounder setting. We visualized some counterfactuals in Fig. H.8. We predict $Scale$ in this setting after we binarize it (large vs. small).

Formally, we quantify model sensitivity to counterfactual changes of a bias variable $X$ via the following measure:

$$
\begin{aligned}
S_X(\theta) := \mathbb{E}_{u \sim U}\, \mathbb{E}_{x \sim \mathrm{Unif}(\mathrm{supp}(X))} \Big[ \\
\big| P(\hat{Scale}(u); \theta_m) - P(\hat{Scale}_x(u); \theta_m) \big| \Big],
\end{aligned}
\tag{20}
$$

and analogously define $S_Y(\theta)$ for the variable $Y$. Intuitively, $S_X$ and $S_Y$ measure the average discrepancy in predictions under counterfactual interventions on $X$ or $Y$, respectively (lower is better). In addition, we report counterfactual accuracy,

$$
\begin{aligned}
Acc_{ctf} := \mathbb{E}_{u \sim U}\, \mathbb{E}_{s \sim \mathrm{Unif}(\mathrm{supp}(Scale))} \Big[ \\
\mathbf{1}\big\{ s = \hat{Scale}_s(u) \big\} \Big],
\end{aligned}
\tag{21}
$$

which evaluates whether predictions remain consistent with ground-truth outcomes under counterfactual changes of the causally relevant target variable $Scale$.

Note that our methods still work solely on observational data. But with the help of this controlled experiment on dSprites, we can even simulate true counterfactuals, and show, that our proposed method actually provides causal stability in terms of counterfactuals.

Counterfactual sensitivity (Tab. 3) reveals that ResNet predictions are highly sensitive to both spurious paths, while

*Table 2.* Performance of all models across 6 datasets. Rows 1–3 (shaded) show naive baselines: *Backbone (Bias)* = lower bound, *Backbone (No Bias)* = upper bound (training data contains no biases, not applicable for TinyBERT and MNLI, see Appendix H.1.6). Regression datasets are reported with $R^2$, classification with Balanced Accuracy or Worst Group Accuracy (WGA) (Appendix H.1.6). Values are mean $\pm$ standard deviation across runs. Train sets are biased (except for No Bias Backbones). Test sets are bias free. †: see Appendix B.

| Model | dSprites ($R^2$) | Blob ($R^2$) | YaleB (BAcc) | FairFace (BAcc) | Waterbirds (BAcc) | MNLI (WGA) |
|---|---|---|---|---|---|---|
| ResNet (Bias) | $0.417 \pm 0.037$ | $0.281 \pm 0.017$ | $0.607 \pm 0.012$ | $0.744 \pm 0.010$ | $0.730 \pm 0.032$ | – |
| ResNet (No Bias) | $0.757 \pm 0.023$ | $0.889 \pm 0.001$ | $0.976 \pm 0.009$ | $0.904 \pm 0.007$ | $0.908 \pm 0.008$ | – |
| TinyBERT (Bias) | – | – | – | – | – | $0.616 \pm 0.010$ |
| $\text{DISCO}_m$ | $\mathbf{0.688 \pm 0.018}$ | $\mathbf{0.759 \pm 0.015}$ | $\mathbf{0.804 \pm 0.027}$ | $\mathbf{0.860 \pm 0.006}$ | $0.867 \pm 0.009$ | $0.751 \pm 0.007$ |
| $\text{sDISCO}^\dagger$ | $0.687 \pm 0.016$ | $0.751 \pm 0.014$ | $0.770 \pm 0.012$ | $0.850 \pm 0.010$ | $0.874 \pm 0.008$ | $\mathbf{0.760 \pm 0.007}$ |
| Adversarial | $0.467 \pm 0.034$ | $0.647 \pm 0.046$ | $0.692 \pm 0.011$ | $0.846 \pm 0.006$ | $0.845 \pm 0.017$ | $0.356 \pm 0.058$ |
| GDRO | – | – | – | $0.840 \pm 0.012$ | $0.865 \pm 0.009$ | $0.755 \pm 0.010$ |
| c-MMD | – | – | – | $0.824 \pm 0.006$ | $0.853 \pm 0.019$ | $0.627 \pm 0.007$ |
| CIRCE | $0.606 \pm 0.021$ | $0.748 \pm 0.015$ | $0.681 \pm 0.026$ | $0.822 \pm 0.006$ | $\mathbf{0.894 \pm 0.010}$ | $0.516 \pm 0.069$ |
| HSCIC | $0.568 \pm 0.033$ | $0.470 \pm 0.022$ | $0.708 \pm 0.035$ | $0.748 \pm 0.013$ | $0.853 \pm 0.012$ | $0.716 \pm 0.020$ |
| IRM | – | – | – | $0.821 \pm 0.017$ | $0.858 \pm 0.009$ | $0.738 \pm 0.008$ |
| Fishr | – | – | – | $0.727 \pm 0.004$ | $0.814 \pm 0.012$ | $0.645 \pm 0.017$ |

$\text{sDISCO}_{XY}$ nearly eliminates such effects and achieves high counterfactual accuracy. Even $\text{sDISCO}_X$, which only observes one of the two biases, substantially reduces sensitivity on both paths, though residual effects remain. This aligns with our theoretical results: unknown biases cannot be removed, but blocking all known biases is the best we can and should do. In the appendix H.4, we further highlight how this pathway analysis can be used to understand why some models failed in learning causally stable relationships in detail in an additional experiment.

## 5. Conclusion

We introduced the Standard Anti-Causal Model (SAM), a unifying framework to analyze bias mechanisms in prediction tasks. From SAM, we derived a conditional independence criterion that guarantees *causal stability*, ensuring predictions rely solely on stable direct effects.

To optimize towards causal stability, we proposed two new estimators of conditional distance correlation: $\text{DISCO}_m$ and sDISCO. While distance correlation had previously been computationally impractical, our methods make it efficient, differentiable, and scalable to deep learning. $\text{DISCO}_m$ provides accurate estimation, while sDISCO offers a single-shot, highly efficient alternative.

Across six datasets with diverse bias structures, $\text{DISCO}_m$ and sDISCO consistently outperform or are competitive against state-of-the-art baselines, while requiring fewer hyperparameters and supporting arbitrary target/bias types. Finally, SAM enables pathway-specific counterfactual analysis, providing deeper insight into model behavior under interventions.

## 6. Limitations and Ethical Considerations

**Theoretical Limitations.** As formally analyzed in Section 2.2, our theoretical guarantees are fundamentally conditional on the assumptions of the Standard Anti-Causal Model (SAM). Violating these core assumptions requires further theoretical or architectural adjustments, as standard application will yield an incomplete mitigation of spurious effects.

**Practical Limitations.** In practice, DISCO estimators operate within the informed bias mitigation regime, requiring that bias variables be explicitly observed and labeled during training. While this is a feasible and standard requirement in some high-stakes domains, such as fairness auditing or medical imaging, it remains a general limitation for broader, unconstrained deployment. Extending bias mitigation efficiently to fully unsupervised scenarios remains an important direction for future work.

**Ethical Considerations.** To evaluate our framework in high-stakes, real-world contexts, we utilized the FairFace dataset (Karkkainen & Joo, 2021) to demonstrate how naive predictors aggressively exploit demographic shortcuts when left unaddressed. In these experiments, we predict "labeled sex," which reflects externally assigned visual categorizations provided by the dataset; we explicitly distinguish this from the distinct, internal, and complex concept of gender. Furthermore, we categorically reject the dataset's original labeling of "race." We view "race" as an ontologically unstable construct that lacks scientific grounding and reproduces historically problematic divides. Because the dataset's "Black" label physically corresponds most consistently to darker skin tones, we reinterpret and relabel this attribute strictly as "skin tone." This physical proxy mapping is not an endorsement of the original taxonomy, but rather a necessary, pragmatic step to effectively isolate and mitigate a highly problematic bias.

## Acknowledgment

This paper is supported by the DAAD programme Konrad Zuse Schools of Excellence in Artificial Intelligence, sponsored by the German Federal Ministry of Research, Technology and Space. We gratefully acknowledge the computational resources provided by the Leibniz Supercomputing Centre.

## Impact Statement

This paper presents work whose goal is to advance the field of machine learning by improving model fairness and robustness against spurious correlations. Because our proposed bias mitigation estimators operate within an informed bias mitigation regime, they require the explicit observation and labeling of bias variables during training. When applied to human-centric domains, this necessitates the careful and ethical categorization of sensitive demographic data. We explicitly address the broader societal consequences and limitations of this requirement, specifically regarding the ontological instability of demographic labels and the distinction between visual categorization and internal identity, within Section 6.

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

# Appendix

## A. Use of LLMs

We used LLMs for this manuscript to polish our writing on a sentence and paragraph level. It was used to intelligently find and eliminate spelling mistakes, improve sentence structure, and to refine the flow of text when necessary.

The used tools are: Grammarly (`https://www.grammarly.com/`), ChatGPT from OpenAI (`https://chatgpt.com/`), and Gemini (`https://gemini.google.com/`).

## B. Camera-Ready Changes

Following the constructive feedback from Reviewer Uimw regarding the presentation of our theoretical claims, we refactored the proofs for our estimators to provide a more direct and rigorous derivation. As detailed in Appendix G, this led to the formalization of the exact algebraic factorization for the sDISCO variant. During this rigorous refactoring process, we discovered a discrepancy in our previous mathematical formulation, which had unfortunately also propagated into our initial code implementation for sDISCO. We have since corrected the implementation to strictly align with the exact mathematical factorization proven in this appendix. The previous results, generated by the prior implementation, are provided in Table 4 for transparency and completeness. However, all main-text results (e.g., Table 2) now reflect the corrected, mathematically faithful sDISCO implementation, which inherently yields stronger empirical performance.

*Table 4.* Old performance table, where sDISCO has implementation errors. Rows 1–3 (shaded) show naive baselines: *Backbone (Bias)* = lower bound, *Backbone (No Bias)* = upper bound (training data contains no biases, not applicable for TinyBERT and MNLI, see Appendix H.1.6). Regression datasets are reported with $R^2$, classification with Balanced Accuracy or Worst Group Accuracy (WGA) (Appendix H.1.6). Values are mean $\pm$ standard deviation across runs. Train set is biased (except for No Bias Backbones). Test set is always bias free.

| Model | dSprites ($R^2$) | Blob ($R^2$) | YaleB (BAcc) | FairFace (BAcc) | Waterbirds (BAcc) | MNLI (WGA) |
|---|---|---|---|---|---|---|
| ResNet (Bias) | $0.417 \pm 0.037$ | $0.281 \pm 0.017$ | $0.607 \pm 0.012$ | $0.744 \pm 0.010$ | $0.730 \pm 0.032$ | – |
| ResNet (No Bias) | $0.757 \pm 0.023$ | $0.889 \pm 0.001$ | $0.976 \pm 0.009$ | $0.904 \pm 0.007$ | $0.908 \pm 0.008$ | – |
| TinyBERT (Bias) | – | – | – | – | – | $0.616 \pm 0.010$ |
| DISCO$_m$ | $\mathbf{0.688 \pm 0.018}$ | $0.759 \pm 0.015$ | $\mathbf{0.804 \pm 0.027}$ | $\mathbf{0.860 \pm 0.006}$ | $0.867 \pm 0.009$ | $\underline{0.751 \pm 0.007}$ |
| sDISCO | $\underline{0.619 \pm 0.011}$ | $\mathbf{0.788 \pm 0.017}$ | $0.694 \pm 0.020$ | $\underline{0.854 \pm 0.005}$ | $0.863 \pm 0.006$ | $0.730 \pm 0.020$ |
| Adversarial | $0.467 \pm 0.034$ | $0.647 \pm 0.046$ | $0.692 \pm 0.011$ | $0.846 \pm 0.006$ | $0.845 \pm 0.017$ | $0.356 \pm 0.058$ |
| GDRO | – | – | – | $0.840 \pm 0.012$ | $0.865 \pm 0.009$ | $\mathbf{0.755 \pm 0.010}$ |
| c-MMD | – | – | – | $0.824 \pm 0.006$ | $0.853 \pm 0.019$ | $0.627 \pm 0.007$ |
| CIRCE | $0.606 \pm 0.021$ | $0.748 \pm 0.015$ | $0.681 \pm 0.026$ | $0.822 \pm 0.006$ | $\mathbf{0.894 \pm 0.010}$ | $0.516 \pm 0.069$ |
| HSCIC | $0.568 \pm 0.033$ | $0.470 \pm 0.022$ | $\underline{0.708 \pm 0.035}$ | $0.748 \pm 0.013$ | $0.853 \pm 0.012$ | $0.716 \pm 0.020$ |
| IRM | – | – | – | $0.821 \pm 0.017$ | $0.858 \pm 0.009$ | $0.738 \pm 0.008$ |
| Fishr | – | – | – | $0.727 \pm 0.004$ | $0.814 \pm 0.012$ | $0.645 \pm 0.017$ |

## C. Contextualization and Related Work

We propose both an analytical causal framework to understand anti-causal prediction from a path-way analysis point of view, and also provide a solution to estimate bias free predictors, as well.

One can thus interpret our work as an candidate of causal representation learning (CRL) methods, or by a causally motivated shortcut/bias mitigation method.

### C.1. SAM and DISCO in Context of CRL

Our work aligns with the central objective of Causal Representation Learning (CRL), which is to learn representations that are robust and generalize across environments by leveraging causal principles. While CRL is a broad field, distinct sub-goals exist, such as: (a) *Generative and Disentangled CRL*, which aims to learn the full causal generative process or disentangle independent causal mechanisms (Sanchez & Tsaftaris, 2022; Komanduri et al., 2024); (b) *Latent Confounder CRL*, which focuses on inferring causal effects when key confounders are unobserved (Wang et al., 2024; Louizos et al., 2017; Kompa et al., 2022); and (c) *Causal Discovery*, which attempts to infer the graph structure from high-dimensional data (Lagemann et al., 2023). These are just very limited sub-fields, we note that there are many more highly relevant CRL subfields (Schölkopf et al., 2021).

Our framework occupies a specific and distinct niche: **Discriminative CRL with observed bias**. Unlike generative approaches, we do not attempt to model the full data-generating process or reconstruct the input. Instead, we demonstrate that for the specific problem of robust prediction, one can avoid the complexity of full generative modeling by adhering to a three-step pipeline:

1. **Modeling:** We propose the Standard Anti-Causal Model (SAM) to formally characterize the data-generating process and its biases $B$.

2. **Criterion:** From SAM, we derive a formal observational criterion, *Causal Stability* ($\hat{Y} \perp B \mid Y$), which we prove is sufficient to ensure that counterfactual indirect and spurious effects vanish, isolating the stable direct effect.

3. **Estimation:** We develop DISCO as an efficient regularization technique to optimize deep models toward this causal criterion.

A key distinction of our approach is the assumption that the bias $B$ is observed. While some CRL methods operate under latent confounding, the assumption of known attributes is foundational to the specific subfield of bias mitigation and is highly realistic in many high-stakes applications. These include fairness auditing (where sensitive attributes like age or sex are collected for compliance), scientific and industrial settings (where metadata regarding sensors or collection times is available), and standard benchmarks explicitly designed to model known spurious correlations (e.g., Waterbirds, FairFace).

### C.2. SAM and DISCO in Context of Shortcut Mitigation

#### C.2.1. SAM

Our work provides a structural foundation that generalizes and clarifies findings from prior influential works in shortcut mitigation, specifically those of Makar et al. (2022); Makar & D'Amour (2023); Veitch et al. (2021); Puli et al. (2021).

**Comparison to Makar et al. (2022); Makar & D'Amour (2023):** Makar & D'Amour (2023) connect risk invariance (a robustness criterion) with separation ($f(X) \perp B \mid Y$) (a fairness criterion). Their analysis operates primarily at the *distributional* level (associated with Rung 1 of Pearl's Ladder of Causation), describing *what* statistical property a robust model should possess. In contrast, SAM operates at the *structural and counterfactual* level (Rung 2 and 3). By making our assumptions explicit via a Structural Causal Model, we analyze the flow of information along specific causal pathways. Since structural guarantees imply distributional ones, our work provides the deeper causal mechanism for *why* the separation principle works: blocking specific spurious paths in the graph necessitates the resulting distributional independence.

**Comparison to Veitch et al. (2021):** Veitch et al. (2021) formalize stress tests using a notion of *unit-level counterfactual invariance*, requiring $f(X(z)) = f(X(z'))$. While theoretically robust, this criterion relies on unobserved potential outcomes and is often impractical to enforce directly. Veitch et al. (2021) show that observational conditional independence is a *necessary* but not *sufficient* signature for this invariance. Our work bridges this gap by defining a more practical notion of invariance tied to observable attributes and path-specific mechanisms within SAM. In our setting, we prove that the conditional independence criterion is *sufficient* to block the transmission of spurious information through specific causal pathways, making the guarantee attainable for practitioners.

**Comparison to Puli et al. (2021):** Puli et al. (2021) (NURD) argue that enforcing conditional independence is too restrictive. They provide examples where an optimal representation $r(x)$ must depend on the nuisance $z$ to maximize mutual information with the label. However, this critique applies to *generative* or *representational* goals, where the aim is to retain as much information about the input $X$ as possible.

1. **Discriminative vs. Generative Targets:** DISCO is designed for a discriminative task. We apply the independence constraint to the *final prediction* $\hat{Y}$, not to an intermediate representation $r(x)$. We allow the model's internal layers to utilize bias information if necessary, provided the final output is purged of spurious signal.

2. **Stability vs. Information:** In the example provided by Puli et al. (2021), the "optimal" representation depends on the nuisance. Consequently, a predictor built on such a representation would also depend on the nuisance. By definition, this results in a predictor that is not causally stable against shifts in that nuisance.

Our objective (maximizing the stable direct effect, $ctf\text{-}DE$) explicitly accepts the trade-off that we do not wish to model the full input $X$. Instead, we isolate the path $Y \to X \to \hat{Y}$ while forcing counterfactual indirect ($ctf\text{-}IE$) and spurious

($ctf$-$SE$) effects to zero. This ensures that the model utilizes the bias $B$ only insofar as it helps extract the stable direct signal, providing a guarantee that is aligned with robust discriminative prediction rather than generative reconstruction.

### C.2.2. DISCO

Finally, within the realm of bias mitigation, we want to further highlight that the causal analysis is not our only contribution. We additionally propose the efficient conditional independence penalty via conditional distance correlation. While we compare against classical baselines such as GDRO (Sagawa et al., 2019), IRM (Arjovsky et al., 2019), Fishr (Rame et al., 2022), and adversarial bias mitigation (Ganin et al., 2016; Wang et al., 2019; Adeli et al., 2021), we especially include the modern approaches based on Reproducing Kernel Hilbert Spaces (RKHs) that also penalize conditional dependence directly. In this category we compare against conditional MMD, called C-MMD (Kaur et al., 2022; Makar & D'Amour, 2023; Veitch et al., 2021), conditional Hilbert-Schmidt Independence Criterion (HSCIC) (Quinzan et al., 2022), and CIRCE (Pogodin et al., 2022) (an efficient variant of HSCIC).

We especially want to highlight that our sDISCO and $DISCO_m$ estimators, together with adversarial bias mitigation, HSCIC and CIRCE, are extremely flexible, as they can be applied to any types of targets and bias attributes, while all the other methods, including the classical gold-standard baselines, always require categorical attributes.

Our methods have significantly less hyperparameters than CIRCE and HSCIC, while being more flexible than the classical methods and consistently providing the best or second best results in our 6 different datasets with different data, bias, and target types.

## D. Causal Graph and Assumptions

### D.1. Valid Adjustment Set

**Characterization of Valid Adjustment Sets (Shpitser et al., 2010)** Let $Z$ be a set of nodes in a causal graph. Then, $Z$ is a valid adjustment set for $(Y, X)$ (implying $Y \cap X = \mathrm{pa}_Y \cap X = \emptyset$) if and only if it satisfies the following conditions:

(i) $Z$ contains no node $R \notin Y$ on a proper causal path from $Y$ to $X$ nor any of its descendants in $G_Y$,

(ii) $Z$ blocks all non-directed paths from $Y$ to $X$.

## E. Causal Invariance as Conditional Independence

In the following, we will prove the propositions in Sec.2.

### E.1. Total Variance Decomposition

**Proposition 2.1 (TV-Decomposition)** The TV can be decomposed into direct, indirect, and spurious components by

$$TV_{y_0,y_1}(\hat{y}) = ctf\text{-}DE_{y_0,y_1}(\hat{y}|y) - ctf\text{-}IE_{y_1,y_0}(\hat{y}|y) - ctf\text{-}SE_{y_1,y_0}(\hat{y}). \tag{22}$$

*Proof.* We first show that $ctf\text{-}TE_{y_0,y_1}(\hat{y}|y_0)$ can be split into its direct and indirect components as given in definition 2.1.

$$ctf\text{-}TE_{y_0,y_1}(\hat{y}|y_0) = P(\hat{y}_{y_1}|y_0) - P(\hat{y}_{y_0}|y_0) \tag{23}$$
$$= P(\hat{y}_{y_1}|y_0) - P(\hat{y}_{y_1,w_{y_0}}|y_0) + P(\hat{y}_{y_1,w_{y_0}}|y_0) - P(\hat{y}_{y_0}|y_0) \tag{24}$$
$$= ctf\text{-}DE_{y_0,y_1}(\hat{y}|y_0) - ctf\text{-}IE_{y_1,y_0}(\hat{y}|y_0). \tag{25}$$

We now can finally show that the total variance can be split into total effects and spurious effects, as given in the following:

$$TV_{y_0,y_1}(\hat{y}) = P(\hat{y}|y_1) - P(\hat{y}|y_0) \tag{26}$$
$$= P(\hat{y}|y_1) - P(\hat{y}_{y_1}|y_0) + P(\hat{y}_{y_1}|y_0) - P(\hat{y}|y_0) \tag{27}$$
$$= P(\hat{y}_{y_1}|y_0) - P(\hat{y}_{y_0}|y_0) + P(\hat{y}_{y_1}|y_1) - P(\hat{y}_{y_1}|y_0) \tag{28}$$
$$= ctf\text{-}TE_{y_0,y_1}(\hat{y}|y_0) - ctf\text{-}SE_{y_1,y_0}(\hat{y}). \tag{29}$$

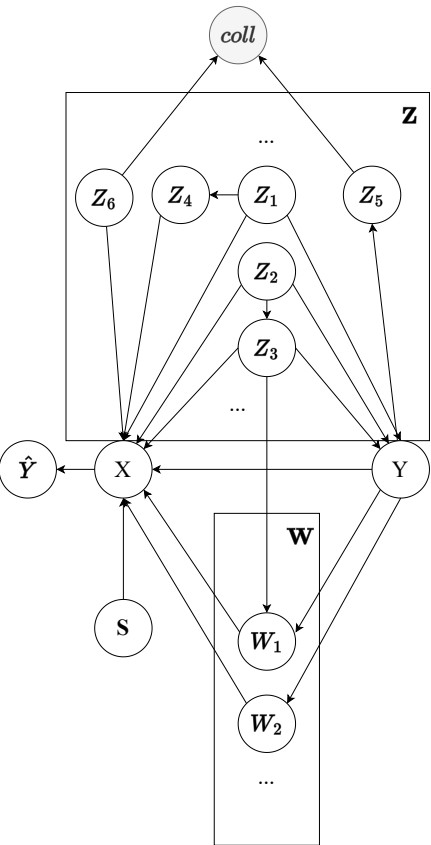

*Figure D.1.* The main setting we consider in anti-causal prediction. Y is the target we want to predict using X only. $\mathbf{W}$ is a set of variables that only permits mediated effects from Y to X. $\mathbf{Z}$ contains nodes that lie on open paths between Y and X but are non-directed, i.e. any kind of nodes that lie on open back-door paths between $X$ and $Y$, or any kinds of paths that are opened due to conditioned colliders. Nodes in $\mathbf{Z}$ are allowed to have arbitrary connections in between them, as long as $\mathbf{Z}$ stays a valid adjustment set for (Y,X), and the graph is still a directed acyclic graph (DAG). Nodes in $\mathbf{W}$ are mediators. Nodes in $\mathbf{S}$ are variables that are d-separation irrelevant between Y and X (and Y and $\hat{Y}$ respectively). We can always ignore the variables in $\mathbf{S}$ for our settings as they do not harm or inform anything for our prediction setup. Thus, these variables are omitted in our analysis and discussions in the main manuscript.

In total, we arrive at

$$TV_{y_0,y_1}(\hat{y}) = ctf\text{-}DE_{y_0,y_1}(\hat{y}|y_0) - ctf\text{-}IE_{y_1,y_0}(\hat{y}|y_0) - ctf\text{-}SE_{y_1,y_0}(\hat{y}). \tag{30}$$

$\square$

**Proposition E.1.** *($ctf\text{-}IE_{y_1,y_0}(\hat{y}|y_0, \mathbf{w}, \mathbf{z})$ is stronger than $ctf\text{-}IE_{y_1,y_0}(\hat{y}|y_0)$) The expression $ctf\text{-}IE_{y_1,y_0}(\hat{y}|y_0, \mathbf{w}, \mathbf{z})$ is a more fine-grained measure of indirect effect than $ctf\text{-}IE_{y_1,y_0}(\hat{y}|y_0)$, as the former captures more nuances of effects than the latter one. Whenever $ctf\text{-}IE_{y_1,y_0}(\hat{y}|y_0, \mathbf{w}, \mathbf{z})$ is zero, $ctf\text{-}IE_{y_1,y_0}(\hat{y}|y_0)$ is zero. The converse is not true.*

*Proof.* We can further extend the total variation formula by the total probability theorem, and we arrive at

$$TV_{y_0,y_1}(\hat{y}) = \sum_{\mathbf{w},\mathbf{z}} ctf\text{-}DE_{y_0,y_1}(\hat{y}|y_0, \mathbf{w}, \mathbf{z})P(\mathbf{w}, \mathbf{z}|y_0) \tag{31}$$

$$- \sum_{\mathbf{w},\mathbf{z}} ctf\text{-}IE_{y_1,y_0}(\hat{y}|y_0, \mathbf{w}, \mathbf{z})P(\mathbf{w}, \mathbf{z}|y_0) \tag{32}$$

$$- ctf\text{-}SE_{y_1,y_0}(\hat{y}) \tag{33}$$

It is therefore easy to see that $ctf\text{-}IE_{y_1,y_0}(\hat{y}|y_0,\mathbf{w},\mathbf{z}) = 0 \implies ctf\text{-}IE_{y_1,y_0}(\hat{y}|y_0) = 0$, but there converse is not true in general. Proof of the falsity of the converse can be given by simple counterexamples. An interested reader can find some in (Plecko & Bareinboim, 2022). $\square$

### E.2. Justification Conditional Independence as Constraint

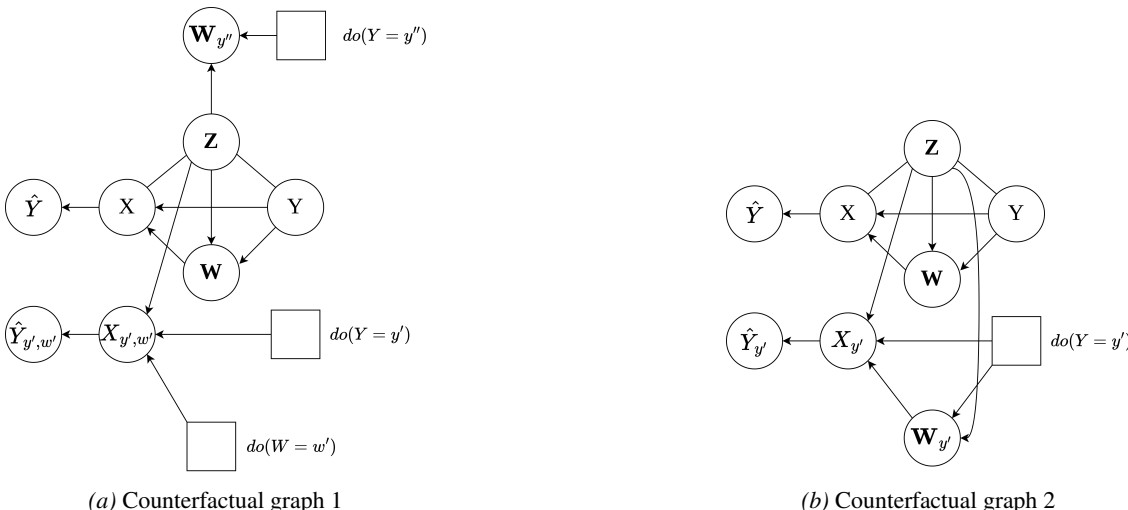

*(a)* Counterfactual graph 1                          *(b)* Counterfactual graph 2

*Figure E.2.* Counterfactual Graphs needed for the proofs of propositions 2 and 3.

**Proposition (Conditional Independence for unbiased Predictors)** SAM is given. Assume we have a prediction model $g_\theta$ that maps from the input $X$ to our target. Let's treat the predictions of our model as a random variable called $\hat{Y}$. If the prediction satisfies $\hat{Y} \perp \mathbf{W}, \mathbf{Z} \mid Y$, then $ctf\text{-}IE_{y_1,y_0}(\hat{y} \mid y) = ctf\text{-}SE_{y_1,y_0}(\hat{y}) = 0$, for any $y, y_0, y_1, \hat{y}$ in their ranges. The conditional independence criterion even ensures a stricter criterion, namely, $\hat{Y} \perp \mathbf{W}, \mathbf{Z} \mid Y \implies ctf\text{-}IE_{y_1,y_0}(\hat{y} \mid y, \mathbf{z}, \mathbf{w}) = 0$, for any $y, y_0, y_1, \hat{y}, \mathbf{w}, \mathbf{z}$ in the ranges of their respective random variables.

Outline

- Prop. E.1 tells us that $ctf\text{-}IE_{y_1,y_0}(\hat{y}|y_0,\mathbf{w},\mathbf{z})$ is stronger than $ctf\text{-}IE_{y_1,y_0}(\hat{y}|y_0)$. Therefore, we will show the sufficiency of the stronger case, which will automatically apply to the weaker one.

- We need to derive some (in)dependence criteria based on the graph. We will use the theory from twin graphs (parallel worlds graphs) (Balke & Pearl, 1994; Avin et al., 2005) and their extensions made in the make-cg algorithm (Shpitser & Pearl, 2007).

- Another useful tool from (Correa et al., 2021) will be used. We will refer to their Theorem 1 as counterfactual unnesting (*ctf-unnesting*) when we use it.

- We first show $\hat{Y} \perp \mathbf{W}, \mathbf{Z}|Y \implies ctf\text{-}IE_{y_1,y_0}(\hat{y}|y,\mathbf{z},\mathbf{w}) = 0$

- We afterwards show $\hat{Y} \perp \mathbf{W}, \mathbf{Z}|Y \implies ctf\text{-}SE_{y_1,y_0}(\hat{y}) = 0$

We begin with the graphical independence criteria. Relevant interventions and observations are in this case $\gamma = (\hat{Y}_{y',\mathbf{w}'}, \mathbf{w}_{y''}, y, \mathbf{w}, \mathbf{z})$. From the graph in Fig. E.2, we can conclude that $\hat{Y}_{y',\mathbf{w}'} \perp \mathbf{W}_{y''}, \mathbf{W}, Y \mid \mathbf{Z}$. Having these relations, we can prove that $\hat{Y} \perp \mathbf{W}, \mathbf{Z}|Y \implies ctf\text{-}IE_{y_1,y_0}(\hat{y}|y,\mathbf{z},\mathbf{w}) = 0$.

*Proof.*

$$
\begin{aligned}
ctf\text{-}IE_{y_1,y_0}(\hat{y}|y,\mathbf{w},\mathbf{z}) &= P(\hat{y}_{y_1,\mathbf{w}_{y_0}}|y,\mathbf{z},\mathbf{w}) - P(\hat{y}_{y_1}|y,\mathbf{z},\mathbf{w}) \\
&= P(\hat{y}_{y_1,\mathbf{w}_{y_0}}|y,\mathbf{z},\mathbf{w}) - P(\hat{y}_{y_1,\mathbf{w}_{y_1}}|y,\mathbf{z},\mathbf{w}) \\
&= \sum_{\mathbf{w}'} P(\hat{y}_{y_1,\mathbf{w}'}, \mathbf{W}_{y_0} = \mathbf{w}'|y,\mathbf{z},\mathbf{w}) \\
&\quad - \sum_{\mathbf{w}''} P(\hat{y}_{y_1,\mathbf{w}''}, \mathbf{W}_{y_1} = \mathbf{w}''|y,\mathbf{z},\mathbf{w}) && \textit{ctf-unnesting} \\
&= \sum_{\mathbf{w}'} P(\hat{y}_{y_1,\mathbf{w}'}|y,\mathbf{z},\mathbf{w}) P(\mathbf{W}_{y_0} = \mathbf{w}'|y,\mathbf{z},\mathbf{w}) \\
&\quad - \sum_{\mathbf{w}''} P(\hat{y}_{y_1,\mathbf{w}''}|y,\mathbf{z},\mathbf{w}) P(\mathbf{W}_{y_1} = \mathbf{w}''|y,\mathbf{z},\mathbf{w}) && \hat{Y}_{y',\mathbf{w}'} \perp \mathbf{W}_{y''} \mid \mathbf{Z} \\
&= \sum_{\mathbf{w}'} P(\hat{y}_{y_1,\mathbf{w}'}|y_1,\mathbf{z},\mathbf{w}') P(\mathbf{w}'|y_0,\mathbf{z}) && \hat{Y}_{y',\mathbf{w}'} \perp Y,\mathbf{W} \mid \mathbf{Z}, \\
&\quad - \sum_{\mathbf{w}''} P(\hat{y}_{y_1,\mathbf{w}''}|y_1,\mathbf{z},\mathbf{w}'') P(\mathbf{w}''|y_1,\mathbf{z}) && \mathbf{W}_{y''} \perp Y,\mathbf{W} \mid \mathbf{Z} \\
&= \sum_{\mathbf{w}'} P(\hat{y}|y_1,\mathbf{z},\mathbf{w}') P(\mathbf{w}'|y_0,\mathbf{z}) \\
&\quad - \sum_{\mathbf{w}''} P(\hat{y}|y_1,\mathbf{z},\mathbf{w}'') P(\mathbf{w}''|y_1,\mathbf{z}) \\
&= P(\hat{y}|y_1) \sum_{\mathbf{w}'} P(\mathbf{w}'|y_0,\mathbf{z}) && \hat{Y} \perp \mathbf{W},\mathbf{Z}|Y \\
&\quad - P(\hat{y}|y_1) \sum_{\mathbf{w}''} P(\mathbf{w}''|y_1,\mathbf{z}) \\
&= 0
\end{aligned}
$$

$\square$

Next, we prove that $\hat{Y} \perp \mathbf{W}, \mathbf{Z}|Y \implies ctf\text{-}SE_{y_1,y_0}(\hat{y}) = 0$.

*Proof.*

$$
\begin{aligned}
SE_{y_1,y_0}(\hat{y}) &= P(\hat{y}_{y_1} \mid y_0) - P(\hat{y}_{y_1} \mid y_1) \\
&= \sum_{\mathbf{z}} [P(\hat{y}_{y_1} \mid y_0,\mathbf{z}) P(\mathbf{z} \mid y_0) - P(\hat{y}_{y_1} \mid y_1,\mathbf{z}) P(\mathbf{z} \mid y_1)] && \text{Law of total Prob.} \\
&= \sum_{\mathbf{z}} [P(\hat{y} \mid y_1,\mathbf{z}) P(\mathbf{z} \mid y_0) - P(\hat{y} \mid y_1,\mathbf{z}) P(\mathbf{z} \mid y_1)] && \hat{Y}_{y_1} \perp Y \mid \mathbf{Z} \\
&= P(\hat{y} \mid y_1) \sum_{\mathbf{z}} P(\mathbf{z} \mid y_0) - P(\hat{y} \mid y_1) \sum_{\mathbf{z}} P(\mathbf{z} \mid y_1) && \hat{Y} \perp \mathbf{W},\mathbf{Z} \mid Y \\
&= 0
\end{aligned}
$$

$\square$

### E.3. Constant Direct Effects given y,y'

**Proposition** (Constant Direct Effect for any $\mathbf{w}, \mathbf{z}$) The graph $\mathcal{G}$ and all implied assumptions are given. Assume we have a predictor $h_{\theta_h} \circ g_{\theta_g}$ and its output represented as a random variable $\hat{Y}$, such that $\hat{Y} \perp \mathbf{W}, \mathbf{Z} \mid Y$ holds. This predictor will have a constant direct effect for fixed $y_0, y_1$, for any $\mathbf{w}, \mathbf{z}$. This is compactly expressed as $ctf\text{-}DE_{y_0,y_1}(\hat{y}|y,\mathbf{w},\mathbf{z}) = ctf\text{-}DE_{y_0,y_1}(\hat{y}|y,\mathbf{w}',\mathbf{z}')$, for any $y_0, y_1, \mathbf{w}, \mathbf{z}, \mathbf{w}', \mathbf{z}'$.

*Proof.*

$$
\begin{aligned}
ctf\text{-}DE_{y_1,y_0}(\hat{y}|y,\mathbf{w},\mathbf{z}) &= P(\hat{y}_{y_1,\mathbf{w}_{y_0}}|y,\mathbf{z},\mathbf{w}) - P(\hat{y}_{y_0}|y,\mathbf{z},\mathbf{w}) \\
&= P(\hat{y}_{y_1,\mathbf{w}_{y_0}}|y,\mathbf{z},\mathbf{w}) - P(\hat{y}_{y_0,\mathbf{w}_{y_0}}|y,\mathbf{z},\mathbf{w}) \\
&= \sum_{\mathbf{w}'}[P(\hat{y}_{y_1,\mathbf{w}'},\mathbf{W}_{y_0}=\mathbf{w}'|y,\mathbf{z},\mathbf{w}) \\
&\quad - P(\hat{y}_{y_0,\mathbf{w}'},\mathbf{W}_{y_0}=\mathbf{w}'|y,\mathbf{z},\mathbf{w})] && \textit{ctf-unnesting} \\
&= \sum_{\mathbf{w}'}[P(\hat{y}_{y_1,\mathbf{w}'}|y,\mathbf{z},\mathbf{w}) && \mathbf{W}_{y''} \perp Y, \mathbf{W} \mid \mathbf{Z} \\
&\quad - P(\hat{y}_{y_0,\mathbf{w}'}|y,\mathbf{z},\mathbf{w})]P(\mathbf{W}_{y_0}=\mathbf{w}'|y,\mathbf{z}) \\
&= \sum_{\mathbf{w}'}[P(\hat{y}|y_1,\mathbf{z},\mathbf{w}') && \hat{Y}_{y',\mathbf{w}'} \perp Y, \mathbf{W} \mid \mathbf{Z} \\
&\quad - P(\hat{y}|y_0,\mathbf{z},\mathbf{w}')]P(\mathbf{W}_{y_0}=\mathbf{w}'|y,\mathbf{z}) \\
&= [P(\hat{y}|y_1) - P(\hat{y}|y_0)]\sum_{\mathbf{w}'}P(\mathbf{W}_{y_0}=\mathbf{w}'|y,\mathbf{z}) && \hat{Y} \perp \mathbf{W}, \mathbf{Z} \mid Y \\
&= P(\hat{y}|y_1) - P(\hat{y}|y_0)
\end{aligned}
$$

$\square$

# F. Proofs and Supplementary Definitions for Conditional Distance Correlation Background

## F.1. Supplementary Definitions

**Definition F.1** (Restatement of Definitions: Conditional distance covariance)**.** Let $P_Z$ denote the probability measure of $Z$. For $z \in \mathcal{Z}$, let $\mu_z := P_{X|Z=z}$ and $\nu_z := P_{Y|Z=z}$ be the respective conditional marginals. For $\theta_z := P_{(X,Y)|Z=z}$, define the distance centered kernels:

$$
d_{\mu_z}(x,x') := d_{\mathcal{X}}(x,x') - a_{\mu_z}(x) - a_{\mu_z}(x') + D(\mu_z),
$$

where $a_{\mu_z}(x) := \int d_{\mathcal{X}}(x,x')\,d\mu_z(x')$, $D(\mu_z) := \int d_{\mathcal{X}}(x,x')\,d\mu_z(x)d\mu_z(x')$, and similarly for $d_{\nu_z}$.

The conditional distance covariance is defined as:

$$
\mathrm{dCov}^2(X,Y \mid Z) := \mathbb{E}_Z\left[\mathrm{dCov}^2(X,Y \mid Z=z)\right],
$$

where for each $z \in \mathcal{Z}$,

$$
\mathrm{dCov}^2(X,Y \mid Z=z) := \int d_{\mu_z}(x,x')\,d_{\nu_z}(y,y')\,dP_{(X,Y)|Z=z}(x,y)dP_{(X,Y)|Z=z}(x',y').
$$

**Definition F.2** (Strong negative type (Lyons, 2013, §3))**.** A metric space $(\mathcal{X}, d)$ has *negative type* if whenever two Borel probability measures $\mu_1, \mu_2$ on $\mathcal{X}$ with finite first moments satisfy

$$
D(\mu_1 - \mu_2) := \iint d(x,x')\,d(\mu_1-\mu_2)(x)\,d(\mu_1-\mu_2)(x') \leq 0.
$$

Here finite first moment means $\int d(o,x)\,d\mu_i(x) < \infty$ for some (hence any) base point $o \in \mathcal{X}$. The same metric space is of *strong negative type* when equality implies $\mu_1 = \mu_2$.

**Definition F.3** (Hilbert embedding (Lyons, 2013, §3))**.** An *isometric embedding* of $(\mathcal{X}, d)$ into a real Hilbert space $\mathcal{H}$ is a map $\varphi : \mathcal{X} \to \mathcal{H}$ such that

$$
d(x,x') = \|\varphi(x) - \varphi(x')\|_{\mathcal{H}}^2 \quad \forall\, x,x' \in \mathcal{X}.
$$

By Schoenberg's theorem (Schoenberg, 1937; 1938), $(\mathcal{X}, d)$ has negative type if and only if such $\varphi$ exists.

**Definition F.4** (Tensor product space)**.** If $\varphi : \mathcal{X} \to \mathcal{H}_X$ and $\psi : \mathcal{Y} \to \mathcal{H}_Y$ are embeddings into real Hilbert spaces, their *tensor-product embedding*

$$
\varphi \otimes \psi \,:\, \mathcal{X} \times \mathcal{Y} \,\to\, \mathcal{H}_X \otimes \mathcal{H}_Y
$$

is defined on simple tensors by

$$(\varphi \otimes \psi)(x, y) \;=\; \varphi(x) \,\otimes\, \psi(y),$$

and extended linearly and by continuity. The inner product on $\mathcal{H}_X \otimes \mathcal{H}_Y$ satisfies

$$\langle u_1 \otimes v_1,\; u_2 \otimes v_2 \rangle = \langle u_1, u_2 \rangle_{\mathcal{H}_X} \,\langle v_1, v_2 \rangle_{\mathcal{H}_Y}.$$

**Definition F.5** (Barycenter map (Lyons, 2013, Prop. 3.1)). Given an embedding $\varphi : \mathcal{X} \to \mathcal{H}$ and any signed Borel measure $\mu$ on $\mathcal{X}$ with finite first moment, define its *barycenter* (or *mean embedding*, compare to maximum mean discrepancy, HSIC, etc., see (Schrab, 2025) as

$$\beta_\varphi(\mu) := \int_{\mathcal{X}} \varphi(x)\, \mu(dx) \;\in \mathcal{H},$$

which is well-defined.

## F.2. Proof of Theorem (Conditional independence iff zero covariance)

*Remark* F.6 (Overloaded notation for $\otimes$). In this setting, the symbol $\otimes$ is used in two distinct contexts, following Lyons' notation (Lyons, 2013). First, for probability measures, as in $\mu_z \otimes \nu_z$, it denotes the *product measure* on the product space $\mathcal{X} \times \mathcal{Y}$. Second, for Hilbert-space-valued embeddings, as in $\varphi(x) \otimes \psi(y)$, it denotes the *tensor product of vectors* in the Hilbert space $\mathcal{H}_X \otimes \mathcal{H}_Y$. The meaning of $\otimes$ must therefore be inferred from context, though its use is unambiguous within the proof structure.

**Theorem F.7** (Restatement of Theorem). *Suppose $(\mathcal{X}, d_{\mathcal{X}})$ and $(\mathcal{Y}, d_{\mathcal{Y}})$ are metric spaces of strong negative type. Let $(X, Y, Z)$ be random elements on a common probability space, and define for each $z$ the conditional measures $\mu_z = P_{X|Z=z}$, $\nu_z = P_{Y|Z=z}$ and $P_{(X,Y)|Z=z}$ as above. Then*

$$\mathrm{dCov}^2(X, Y \mid Z) = 0 \quad \Longleftrightarrow \quad P_{(X,Y)|Z=z} \;=\; \mu_z \otimes \nu_z \quad \textit{for } P_Z\textit{-a.e. } z.$$

*Proof.* Choose isometric embeddings

$$\varphi : \mathcal{X} \hookrightarrow \mathcal{H}_X, \qquad \psi : \mathcal{Y} \hookrightarrow \mathcal{H}_Y$$

into real Hilbert spaces. Thereby automatically follows that each of $\mathcal{X}, \mathcal{Y}$ has strong negative type, and moreover so that each barycenter map $\beta_\varphi, \beta_\psi$ is injective on all finite-moment signed measures (Lyons, 2013, Lem. 3.9). Consider the tensor-product embedding

$$\varphi \otimes \psi : \mathcal{X} \times \mathcal{Y} \to \mathcal{H}_X \otimes \mathcal{H}_Y,$$

and let $\theta_z := P_{(X,Y)|Z=z}$ be the conditional joint law on $\mathcal{X} \times \mathcal{Y}$, with marginals $\mu_z, \nu_z$.

Analog to Lyons (2013, Prop. 3.7), we have for each $z$ the identity

$$\mathrm{dCov}^2(X, Y \mid Z = z) \;=\; 4 \left\| \beta_{\varphi \otimes \psi} \big( \theta_z - \mu_z \otimes \nu_z \big) \right\|^2_{\mathcal{H}_X \otimes \mathcal{H}_Y}.$$

Taking expectation in $z \sim P_Z$ gives

$$\mathrm{dCov}^2(X, Y \mid Z) = \mathbb{E}_Z \,\mathrm{dCov}^2\big(X, Y \mid Z = z\big) = 4\, \mathbb{E}_Z \left\| \beta_{\varphi \otimes \psi}(\theta_z - \mu_z \otimes \nu_z) \right\|^2.$$

Since norms are nonnegative and barycenters are well-defined on all finite-moment signed measures, it follows that

$$\mathrm{dCov}^2(X, Y \mid Z) = 0 \quad \Longleftrightarrow \quad \beta_{\varphi \otimes \psi}(\theta_z - \mu_z \otimes \nu_z) = 0 \quad P_Z\text{-a.s.}$$

Injectivity of $\beta_{\varphi \otimes \psi}$ (Lyons, 2013, Lem. 3.9) on signed measures then yields

$$\theta_z - \mu_z \otimes \nu_z = 0 \quad \Longleftrightarrow \quad \theta_z = \mu_z \otimes \nu_z,$$

for $P_Z$-almost every $z$, as required. $\qquad\qquad\square$

## F.3. Consistency of DISCO$_m$

**Proposition F.8** (Consistency of DISCO$_m$). *Let $\{(X_i, Y_i, Z_i)\}_{i=1}^n \overset{i.i.d.}{\sim} P_{XYZ}$ with finite first moments. Let $K_h$ be a positive-definite kernel on $\mathcal{Z}$ with bandwidth $h$, such that standard kernel regression assumptions hold:*

- *$K$ is bounded, continuous, and integrates to $1$.*

- *$h \to 0$ and $nh^{d_Z} \to \infty$ as $n \to \infty$.*

*Then, as $n \to \infty$ and $m \to \infty$ (where $m \leq n$), the estimator converges in probability to the true conditional distance correlation:*

$$DISCO_m(X, Y \mid Z) \overset{p}{\to} \mathrm{dCor}^2(X, Y \mid Z). \tag{34}$$

*Proof.* **Step 1. Pointwise Consistency of the Local Estimator.**
For a fixed reference point $Z_i$, the row-normalized kernel weights $w_k^{(i)} = K_h(Z_i, Z_k)/\sum_\ell K_h(Z_i, Z_\ell)$ form the classical Nadaraya-Watson estimator of the conditional density at $Z_i$. As established by Wang et al. (2015), applying these weights to the double-centered distance matrices $A^{(i)}$ and $B^{(i)}$ constitutes a plug-in empirical estimator for the local V-statistic. Under the stated bandwidth conditions, this local estimator converges in probability to the true local conditional distance covariance:

$$\widehat{\mathrm{dCov}}_{\mathrm{DISCO}}(X, Y \mid Z_i) \overset{p}{\to} \mathrm{dCov}^2(X, Y \mid Z = Z_i). \tag{35}$$

**Step 2. Consistency of the Local Variances.**
By symmetry, substituting $Y$ for $X$ or $X$ for $Y$ in the local estimator yields the local conditional distance variances. Consequently, we have:

$$\widehat{\mathrm{dCov}}_{\mathrm{DISCO}}(X, X \mid Z_i) \overset{p}{\to} \mathrm{dVar}^2(X \mid Z = Z_i), \tag{36}$$

$$\widehat{\mathrm{dCov}}_{\mathrm{DISCO}}(Y, Y \mid Z_i) \overset{p}{\to} \mathrm{dVar}^2(Y \mid Z = Z_i). \tag{37}$$

**Step 3. Convergence of the Global Expectation.**
The theoretical global conditional distance correlation is defined as the expected value of the local correlations over the marginal distribution of $Z$:

$$\mathrm{dCor}^2(X, Y \mid Z) = \mathbb{E}_Z \left[ \frac{\mathrm{dCov}^2(X, Y \mid Z = z)}{\sqrt{\mathrm{dVar}^2(X \mid Z = z)\, \mathrm{dVar}^2(Y \mid Z = z)}} \right]. \tag{38}$$

The DISCO$_m$ estimator approximates this global expectation by uniformly sampling $m$ reference points from the empirical distribution of $Z$ and computing the empirical mean of the local plug-in estimators:

$$\mathrm{DISCO}_m(X, Y \mid Z) = \frac{1}{m} \sum_{i=1}^m \frac{\widehat{\mathrm{dCov}}_{\mathrm{DISCO}}(X, Y \mid Z_i)}{\sqrt{\widehat{\mathrm{dCov}}_{\mathrm{DISCO}}(X, X \mid Z_i)\, \widehat{\mathrm{dCov}}_{\mathrm{DISCO}}(Y, Y \mid Z_i)}}. \tag{39}$$

**Step 4. Ratio Convergence and the Law of Large Numbers.**
Because the numerator and both components of the denominator inside the sum are consistent estimators of their respective population limits (Steps 1 and 2), the continuous mapping theorem (Slutsky's theorem) guarantees that their ratio converges in probability to the true local correlation for any valid $Z_i$.

Finally, as $m \to \infty$, the empirical average over the $m$ sampled points converges in probability to the true expectation over the marginal distribution of $Z$ by the Weak Law of Large Numbers. Therefore, the global estimator converges to the theoretical global conditional distance correlation. $\square$

# G. Exact Factorization and Proof of sDISCO

In Section 3.1, we established that the standard evaluation of the conditional distance correlation V-statistic requires computing localized double-centered matrices for each conditioning point. Naively parallelizing this for a batch of size $n$ requires allocating a tensor $\mathcal{T} \in \mathbb{R}^{n \times n \times n}$, yielding an $\mathcal{O}(n^3)$ memory complexity that is prohibitive for deep learning hardware. Here, we provide the structural proof for the Single-Shot DISCO (sDISCO) estimator. We prove that sDISCO achieves the exact mathematical evaluation of the global V-statistic without ever materializing an $\mathcal{O}(n^3)$ tensor, strictly confining memory requirements to $\mathcal{O}(n^2)$ through exact algebraic factorization.

**Theorem G.1** (Algebraic Factorization of Conditional Distance Covariance). *Let $A, B \in \mathbb{R}^{n \times n}$ be the pairwise distance matrices for variables $X$ and $Y$. Let $W \in \mathbb{R}^{n \times n}$ be the row-normalized kernel weight matrix, where $W_{ij} = K_h(Z_i, Z_j) / \sum_k K_h(Z_i, Z_k)$. Let $\mathbf{1} \in \mathbb{R}^n$ be a column vector of ones, and let $\circ$ denote the Hadamard (element-wise) product.*

*The vector of exact local sample conditional distance covariances $\mathcal{V}_{XY} \in \mathbb{R}^n$, evaluated at all $n$ reference points simultaneously, is given exactly by:*

$$\mathcal{V}_{XY} = T_1 + T_2 - 2T_3, \tag{40}$$

*where the components are derived purely via $\mathcal{O}(n^2)$ matrix multiplications:*

$$T_1 = (W \circ (W(A \circ B)))\mathbf{1}, \tag{41}$$
$$T_2 = ((W \circ (WA))\mathbf{1}) \circ ((W \circ (WB))\mathbf{1}), \tag{42}$$
$$T_3 = (W \circ (WA) \circ (WB))\mathbf{1}. \tag{43}$$

*Proof.* Consider a fixed reference point $Z_i$ corresponding to the $i$-th row of the weight matrix, denoted as the weight vector $w^{(i)} \in \mathbb{R}^n$. Following the V-statistic formulation (Wang et al., 2015), the exact sample conditional distance covariance at $Z_i$ can be expanded into three components: $\mathcal{V}_{XY}^{(i)} = D_1^{(i)} + D_2^{(i)} - 2D_3^{(i)}$, where:

$$D_1^{(i)} = \sum_{k=1}^{n} \sum_{l=1}^{n} w_k^{(i)} w_l^{(i)} A_{kl} B_{kl}, \tag{44}$$

$$D_2^{(i)} = \left( \sum_{k=1}^{n} \sum_{l=1}^{n} w_k^{(i)} w_l^{(i)} A_{kl} \right) \left( \sum_{k=1}^{n} \sum_{l=1}^{n} w_k^{(i)} w_l^{(i)} B_{kl} \right), \tag{45}$$

$$D_3^{(i)} = \sum_{k=1}^{n} \sum_{l=1}^{n} \sum_{m=1}^{n} w_k^{(i)} w_l^{(i)} w_m^{(i)} A_{kl} B_{km}. \tag{46}$$

To vectorize this over all $i \in \{1 \dots n\}$ without $\mathcal{O}(n^3)$ memory, we rely on the Diagonal Extraction Identity for quadratic forms. For any matrix $S \in \mathbb{R}^{n \times n}$, the diagonal of $WSW^\top$ computes the weighted sum $w^{(i)} S w^{(i)\top}$ for all rows $i$. We can extract this diagonal exactly without computing the off-diagonal elements via:

$$\mathrm{diag}(WSW^\top) = (W \circ (WS))\mathbf{1}. \tag{47}$$

We systematically apply this identity to factorize the three V-statistic components:

**1. The Joint Distance Term ($T_1$):**
Let $S = A \circ B$. For a single reference point $i$, $D_1^{(i)} = w^{(i)} S w^{(i)\top}$. Using the identity from Eq. 47, we compute this for all $n$ points simultaneously:

$$T_1 = (W \circ (W(A \circ B)))\mathbf{1}. \tag{48}$$

**2. The Grand Mean Term ($T_2$):**
The term $D_2^{(i)}$ is the product of the weighted grand means of $X$ and $Y$. The grand mean for $X$ at reference point $i$ is exactly $w^{(i)} A w^{(i)\top}$. Applying Eq. 47, the vector of all grand means for $X$ is $g^X = (W \circ (WA))\mathbf{1}$. Thus, the vectorized component $T_2$ is simply the Hadamard product of the grand mean vectors:

$$T_2 = g^X \circ g^Y. \tag{49}$$

**3. The Cross Term** ($T_3$)**:**

To compute $D_3^{(i)}$, we first factor the summation to isolate the local row means. Let $M^X = WA$ and $M^Y = WB$ be the matrices of local row means, such that $M_{ik}^X = \sum_{l=1}^n W_{il} A_{kl}$ (since $A$ is symmetric). We can rewrite $D_3^{(i)}$ as:

$$D_3^{(i)} = \sum_{k=1}^n W_{ik} \left( \sum_{l=1}^n W_{il} A_{kl} \right) \left( \sum_{m=1}^n W_{im} B_{km} \right) = \sum_{k=1}^n W_{ik} M_{ik}^X M_{ik}^Y. \tag{50}$$

Vectorizing this row-wise dot product for all $n$ points yields:

$$T_3 = (W \circ M^X \circ M^Y)\mathbf{1}. \tag{51}$$

By substituting $T_1$, $T_2$, and $T_3$, we perfectly reconstruct $\mathcal{V}_{XY} = T_1 + T_2 - 2T_3$.

**Complexity Analysis:** The naïve execution of the double-centered matrix expansions (as required by the exact definition of distance correlation) necessitates intermediate storage of size $\mathcal{O}(c \cdot n^2)$ for $c$ reference points. For a fully global estimator ($c = n$), this scales as $\mathcal{O}(n^3)$, which causes Out-Of-Memory errors on modern GPUs for standard batch sizes. The sDISCO factorization avoids this entirely. The densest operations are standard matrix multiplications of size $(n \times n) \times (n \times n)$, such as $WA$ and $W(A \circ B)$. The Hadamard product acts as a computational filter, reducing the implicit tensor contraction to a row-wise summation before any third-order dimension is ever instantiated. Consequently, sDISCO leverages the $\mathcal{O}(n^3)$ compute capacity of highly optimized GPU Tensor Cores while remaining strictly bounded by an $\mathcal{O}(n^2)$ memory footprint. $\qquad\square$

# H. Experiments

## H.1. Datasets

### H.1.1. CUSTOM DSPRITES

We implemented our own version of the dSprites dataset (Matthey et al., 2017) that allows for full control of the shapes. We control the shape, orientation, x/y-positions, and the scale of objects placed within a 2D grid.

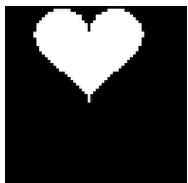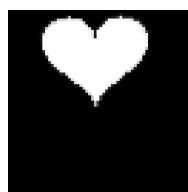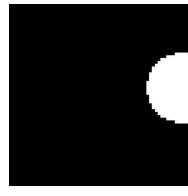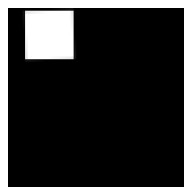

*Figure H.3.* Custom dSprites examples.

Based on Matthey et al. (2017), we construct images of geometric shapes (Sh) with different orientation (O), scale (Sc), y-position (Y), and x-position (X). The regression target, Y, nonlinearly determines the object's y-position while X acts as a confounder. A biased predictor will use both spatial dimensions, while a robust predictor will solely/mainly rely on the y-axis. The SCM is:

$$U_x \sim \text{Uniform}\left(0, \tfrac{\pi}{2}\right), \quad U_y \sim \mathcal{N}\left(0, 0.15^2\right),$$
$$U_{sc} \sim \text{Uniform}(0.5, 0.7), \quad U_\theta \sim \text{Uniform}(0, 360°),$$
$$U_{\text{shape}} \sim \text{UniformDiscrete}\{\text{square}, \text{ellipse}, \text{heart}\},$$
$$\varepsilon_x \sim \mathcal{N}\left(0, 0.01^2\right), \quad \varepsilon_{y1} \sim \mathcal{N}\left(0, 0.1^2\right),$$
$$\varepsilon_{y2} \sim \mathcal{N}\left(0, 0.2^2\right), \quad x = \sin(U_x), \quad y = x^2 + U_y,$$
$$\text{Im} := f\big(x + \varepsilon_x, \ \exp(y + \varepsilon_{y1}) + \varepsilon_{y2}, U_{scale}, U_{shape}, U_\theta\big)$$

### H.1.2. BLOB DATASET

Inspired by Adeli et al. (2021), we generate synthetic images with two Gaussian blobs. We control the intensities of the blobs, and bias their relationships as described below.

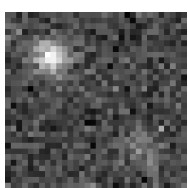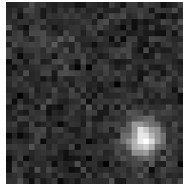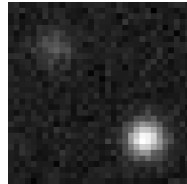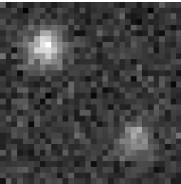

*Figure H.4.* Blob dataset examples.

One blob's intensity is causally related to the regression target, causal intensity (CI), while the other acts as a spurious mediator, bias intensity (BI). A bias-free predictor will only rely on the upper left blob, while a bias predictor will base its predictions on both blobs. The equations of the simulator are:

$$U_{\text{causal}} \sim \text{Unif}(0, 1), \quad U_{\text{bias}} \sim \mathcal{N}(0, 0.1^2),$$
$$\varepsilon_{\text{causal}} \sim \mathcal{N}(0, 0.1^2), \quad CI := U_{\text{causal}},$$
$$BI := CI + U_{\text{bias}},$$
$$Im := f(\exp(\text{CI} + \varepsilon_{\text{causal}}), \exp(\text{BI})).$$

### H.1.3. YALEB.

The extended YaleB dataset (Georghiades et al., 2001; Yale, 2001) gives us images of faces in different poses. Additionally, these images have a single light source that is varied across different combinations. We utilize the azimuth and elevation of the light source as a bias as described in the main body of this manuscript.

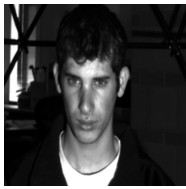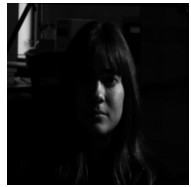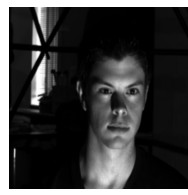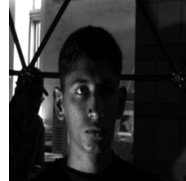

*Figure H.5.* Extended yaleB examples.

From the extended YaleB dataset (Georghiades et al., 2001; Yale, 2001), we predict face pose (collapsed into three categories: frontal, slightly left, maximally left). Azimuth (Az) and elevation (El) of the light source serve as continuous bias variables. To introduce realistic bias settings in YaleB, we used selection bias to correlate the target labels with lighting directions:

$$X = (\text{azimuth}, \text{elevation}), \quad v = (1, 1)^\top$$
$$s = v^\top \, \text{Standardize}(X),$$
$$z = \text{QuantilePartition}(s, 3) \in \{0, 1, 2\},$$
$$\Pr(S = 1 \mid z, \text{pose}) = \begin{cases} 1.0 & \text{if pose} = z, \\ 0.05 & \text{if pose} \neq z. \end{cases}$$

### H.1.4. FAIRFACE

The FairFace dataset (Karkkainen & Joo, 2021) provides a range of demographic annotations that can be leveraged to study and mitigate bias. In our experiments, we focus on sex and skin tone as salient attributes. Since FairFace does not provide an explicit skin tone annotation, we inferred it by reinterpreting the dataset's "race" attribute, specifically the "Black" label, which in practice corresponds most consistently to darker skin tones. Importantly, we do not endorse the conceptualization of "race" as used in FairFace: the category is ontologically unstable, lacks scientific grounding, and reproduces historically problematic constructs.

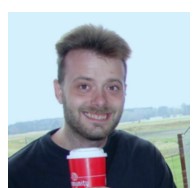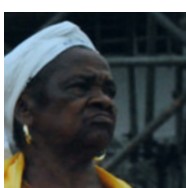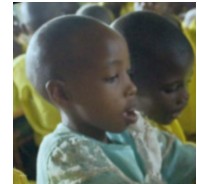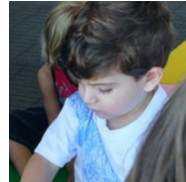

*Figure H.6.* FairFace examples.

We use FairFace (Karkkainen & Joo, 2021) to predict sex as the target. Skin tone (light vs. dark) acts as a bias through selection bias. If formalized as a SCM, the process would follow:

$$P(S = 1 \mid Y, B) = \begin{cases} 0.9 & \text{if } Y = B \text{ (aligned)}, \\ 0.1 & \text{if } Y \neq B \text{ (misaligned)}. \end{cases}$$

### H.1.5. WATERBIRDS

We adopt the Waterbirds dataset (Sagawa et al., 2019), constructed by overlaying birds (Welinder et al., 2010; Wah et al., 2011) on land or water backgrounds (Zhou et al., 2017).

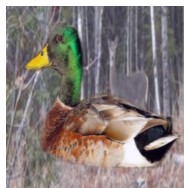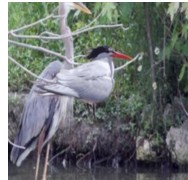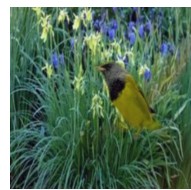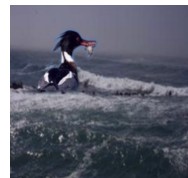

*Figure H.7.* Waterbirds examples.

For Waterbirds, we have again access to the data generating process. The recipe is to select the backgrounds for each bird, to introduce a heavy shortcut. The background is a mediator variable, and the formal SCM reads:

$$bird \sim \text{Bern}(0.5),$$
$$background \mid bird \sim \text{Bern}\big(0.9 \cdot bird + 0.1 \cdot (1 - bird)\big)$$

### H.1.6. MNLI

**Dataset and Experimental Setup.** We use the MNLI dataset (Williams et al., 2018), adopting the exact bias mitigation setup described by Sagawa et al. (2019). The task is to predict the relationship $\in \{entailment, neutral, contradiction\}$ between a sentence pair, where negation terms (e.g., 'nobody', 'no', 'never', 'nothing') act as spurious features highly correlated with the target label.

Unlike our other datasets (Waterbirds, dSprites, Blob, FairFace, YaleB), which offered enough control or balance to construct unbiased validation and test sets, the MNLI base dataset is heavily biased. Since creating a strictly unbiased test set would require discarding excessive data, we follow the protocol of (Sagawa et al., 2019): we report Worst Group Accuracy (WGA) and perform model selection based on this metric. Note that WGA is only applicable to datasets where both targets and bias attributes are **categorical**.

### H.2. Bias Mitigation Methods

All of the baseline methods apply different ideas and techniques, but need a backbone architecture. We used the same architecture, depending on the dataset, for all bias mitigation methods. We utilized the classical ResNet (He et al., 2016) in all experiments. To be more detailed, we used the ResNet-18 variant implemented in the torchvision library (TorchVision maintainers and contributors, 2016), pretrained on ImageNet (IMAGENET1K_V1)(Deng et al., 2009), for dSprites, yaleB, Waterbirds, and FairFace. For the Blob dataset, which deals with smaller images, we used our own implementation of the ResNet architecture that uses group norm (GN) instead of batch norm, as GN was more stable in our experiments. We also changed the depth to two residual blocks, and edited the maxpooling logic to fit the requirements for this small resolution dataset.

For MNLI, we used the original TinyBERT of Jiao et al. (2020) and directly used their officially pre-trained version from HuggingFace at `https://huggingface.co/huawei-noah/TinyBERT_General_4L_312D`.

All details can be found in our repository.

### H.2.1. GDRO

We implemented GDRO exactly following the code and paper by Sagawa et al. (2019). GDRO is one of the standard methods when it comes to bias mitigation as it directly builds on the DRO method that can be seen as a defacto standard in domain generalization (Ben-Tal & Nemirovski, 2002).

### H.2.2. ADVERSARIAL

There are many different suggestions how to design adversarial networks for domain generalization or bias mitigation (Ganin et al., 2016; Wang et al., 2019; Adeli et al., 2021). But most of them differ in some details. The adversarial network, as all other methods in this paper as well, uses the ResNet backbone. Following the ResNet activations, we attach different heads. One head will always be responsible for the task prediction, thus predicting the target $Y$. Then, for every bias variable present, a new head will be introduced. The optimization procedure is as follows:

- predict the task. Update ResNet and the task head parameters.

- for each bias, predict it. Update only the bias prediction heads (protected attribute step).

- for each bias, predict it. Use the negative gradient of the loss and update only the ResNet (unlearn step).

There are many different ways to parametrize the optimization procedure for this adversarial setting. Some suggest using different learning rates for each of the network parts, including each of the bias heads. We, instead, use a single learning rate for the entire network with all of its parts. To adjust the weighting of the different network parts, we use $\lambda_{protected}$ as the parameter that controls the learning weight for the task head. All bias heads use the same weighting parameter $\lambda_{unlearn}$. Instead of using different learning rates, this loss weighting reduces to an equivalent formulation, while being more comparable to the other, regularzation-based methods. See Tab. 5 for the search space.

### H.2.3. C-MMD

We implemented c-MMD, as formulated similarly by Makar et al. (2022); Kaur et al. (2022); Veitch et al. (2021). It tries to penalize predictions or latent representations based on conditional MMD. Besides the penalty strength $\lambda$, it introdcues a single bandwidth (bw) hyperparameter that controls the (Gaussian) kernel's bandwidth on the network prediction outputs. See Tab. 5 for the search space.

### H.2.4. HSCIC

This method is an direct extension of the classical HSIC measure (Gretton et al., 2005; 2007) to the conditional setting. Quinzan et al. (2022) first used it in their work, but their implementation is extremely inefficient. We directly adapted the efficient implementation given by Pogodin et al. (2022).

This method is the most expensive one, when it comes to hyperparameters. It introduces 5 hyperparameters. We did a grid search over 4 of them, as this space was already immense, compared to the others, and held one of them fixed, based on initial results. The 4 hyperparameters we searched over is the regularization strength, $\lambda$, and three bandwidth parameters, $\sigma$, for each of the involved variables in the conditional independence criterion. It further has another parameter, the ridge regression regularization parameter, $\lambda_{ridge}$, we which held fixed at 0.01, following similar values as in Pogodin et al. (2022); Quinzan et al. (2022), and also ensuring in preliminary runs that this value is advantageous for the method in our settings. See Tab. 5 for the search space.

### H.2.5. CIRCE

CIRCE (Pogodin et al., 2022) is a direct extension of the HSCIC method. It was published as a direct successor or competitor. CIRCE, in theory, has a similar set of hyperparameters. But it reduces the search space by doing a small, pre-training hyperparameter optimization that does not depend on the neural network, in advance. See Tab. 5 for the search space.

### H.2.6. DISCO

Our methods, DISCO$_m$ and sDISCO, introduce the regularization strength $\lambda$ and the bandwidth (bw) parameter for the Gaussian kernel. See Tab. 5 for the search space.

### H.2.7. IRM

Invariant Risk Minimization (IRM) (Arjovsky et al., 2019) is a prominent approach from the domain generalization literature designed to learn representations that elicit invariant optimal predictors across multiple environments. In our bias mitigation framework, however, we do not have predefined environments in the traditional sense, and thus we do not perform environment-stratified sampling. Instead, we use standard random batch sampling, consistent with all other baseline methods, and adapt the IRM penalty to operate on a group-based logic, treating the combinations of targets and biases as groups (similar to the GDRO setup). We compute the IRM penalty across these discrete groups rather than distinct training environments. See Tab. 5 for the search space, which includes the penalty weight $\lambda$ and a penalty annealing/warm-up period.

### H.2.8. FISHR

Fishr (Rame et al., 2022) is another method rooted in domain generalization that seeks to align the variance of gradients across different environments to promote out-of-distribution generalization. Similar to IRM, Fishr natively assumes the availability of discrete training environments. Because our setup evaluates models on strictly biased datasets without explicit environment partitions, we bypass environment-balanced sampling and maintain standard random sampling. We adapt Fishr's core logic to our framework by computing and regularizing the gradient variances across our defined demographic or target-bias groups instead. See Tab. 5 for the search space, which similarly includes the regularization strength $\lambda$ and the warm-up epochs.

### H.2.9. HYPERPARAMETERS

*Table 5.* Hyperparameter grids searched for each method, along with the total number of runs evaluated.

| Method | Hyperparameter Grid | Runs |
|---|---|---|
| Adversarial Network | $\lambda_{\text{protected}} \in \{10.0, 5.0, 2.0, 1.0, 0.5, 0.1\}$ 
 $\lambda_{\text{unlearn}} \in \{10.0, 5.0, 2.0, 1.0, 0.5, 0.1\}$ | $6 \times 6 = 36$ |
| DISCO (both) | $\text{bw} \in \{1.0, 0.9, 0.5, 0.1, 0.01, 0.001\}$ 
 $\lambda \in \{10.0, 5.0, 2.0, 1.0, 0.5, 0.1\}$ | $6 \times 6 = 36$ |
| CIRCE | $\lambda \in \{1000.0, 100.0, 10.0, 1.0, 0.1\}$ 
 $\sigma_{\hat{y}}^2 \in \{0.9, 0.5, 0.1, 0.01, 0.001\}$ 
 $\sigma_z^2 \in \{0.9, 0.5, 0.1, 0.01, 0.001\}$ | $5 \times 5 \times 5 = 125$ |
| c-MMD | $\text{bw} \in \{1.0, 0.9, 0.5, 0.1, 0.01, 0.001, 0.0001, 0.00001\}$ 
 $\lambda \in \{1000.0, 100.0, 10.0, 1.0, 0.5, 0.1\}$ | $8 \times 6 = 48$ |
| HSCIC | $\lambda \in \{1000.0, 100.0, 10.0\}$ 
 $\sigma_{\hat{y}}^2 \in \{0.5, 0.1, 0.01, 0.001\}$ 
 $\sigma_z^2 \in \{0.1, 0.01, 0.001\}$ 
 $\sigma_y^2 \in \{0.1, 0.01, 0.001\}$ | $3 \times 4 \times 3 \times 3 = 108$ |
| IRM | $\lambda \in \{1.0, 2.0, 10.0, 100.0, 500.0, 1000.0, 5000.0, 10000.0, 50000.0, 100000.0\}$ 
 warm-up epochs $\in \{0, 10, 30, 40\}$ | $10 \times 4 = 40$ |
| Fishr | $\lambda \in \{1.0, 2.0, 10.0, 100.0, 500.0, 1000.0, 5000.0, 10000.0, 50000.0, 100000.0\}$ 
 warm-up epochs $\in \{0, 10, 30, 40\}$ | $10 \times 4 = 40$ |

## H.3. Path Analysis Counterfactuals

As a recap, for this experiment, scale is our target attribute, and we predict it in a binary manner (large vs. small scale). The $X$ and $Y$ positions of the object itself work as biases we want to mitigate.

In the manuscript, we quantified model sensitivity to counterfactual changes of a bias variable $X$ via the following measure:

$$S_X(\theta) := \mathbb{E}_{u \sim U} \, \mathbb{E}_{x \sim \text{Unif}(\text{supp}(X))} \left[ \left| P(\hat{Y}(u); \theta_m) - P(\hat{Y}_x(u); \theta_m) \right| \right],$$

and analogously defined $S_Y(\theta)$ for the variable $Y$.

In addition, we further introduced counterfactual accuracy,

$$Acc_{ctf} := \mathbb{E}_{u \sim U} \, \mathbb{E}_{s \sim \text{Unif}(\text{supp}(Scale))} \left[ \mathbf{1}\big\{Y(u) = \hat{Y}_s(u)\big\} \right],$$

which evaluates whether predictions remain consistent with ground-truth outcomes under counterfactual changes of the causally relevant target variable Scale.

To visually understand what these formulas measure, we fix a single unit $u \in U$. The following Fig. H.8 shows what some of the different counterfactuals look like for this given unit $u$.

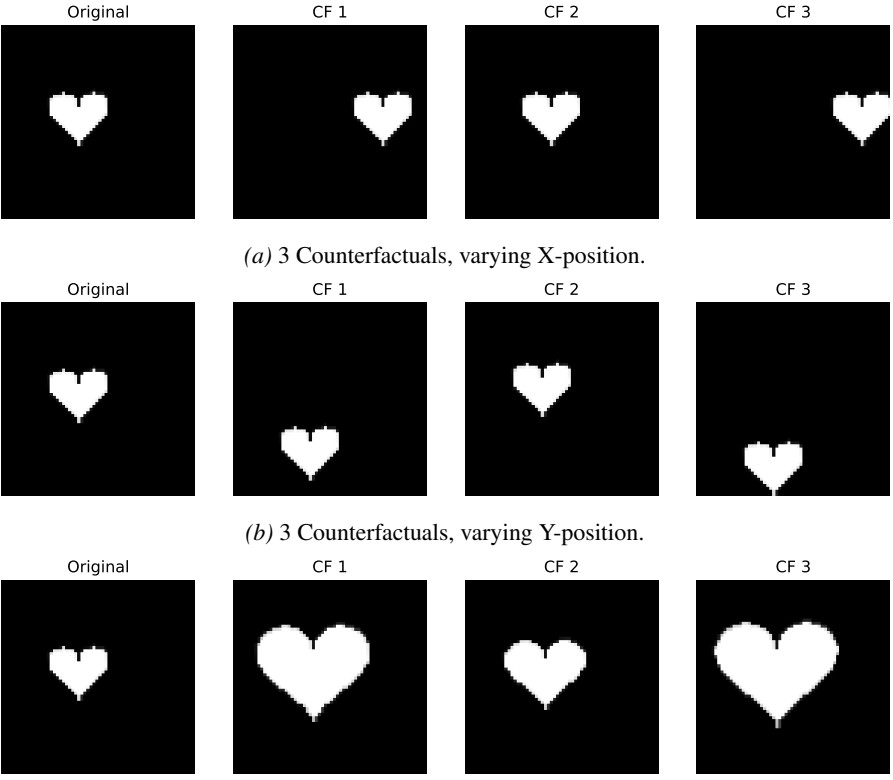

*(a)* 3 Counterfactuals, varying X-position.

*(b)* 3 Counterfactuals, varying Y-position.

*(c)* 3 Counterfactuals, varying Scale.

*Figure H.8.* Counterfactual images for a given unit $u$, labeled here as "Original".

### H.4. Counterfactual Analysis on dSprites

To further illustrate the benefits of counterfactual analysis, we consider the same dSprites setup as in the main manuscript (Sec. 4). Specifically, we evaluate one of the already hyperparameter-optimized, best-performing variants for each method.

In the main experiments, we followed standard practice in the domain generalization literature and reported results on bias-balanced validation and test sets. While this setting is appropriate for assessing overall causal stability in a single balanced setting, it does not necessarily reveal *why* a model fails. In particular, causal stability is not the only goal. While a model might be stable to counterfactual changes of the bias attribute, it might do so by also reducing overall predictive performance.

Counterfactual analysis, grounded in the causal graph formalized through SAM, provides these exact complementary insights. To recall the setup from the Section 4: the target variable $Y$ is the vertical position (y-coordinate) of the object in the 2D plane, while the bias attribute is the horizontal position (x-coordinate), denoted as $X$.

Analogous to pathway analysis, we define

$$S_X(\theta) := \mathbb{E}_{u \sim U} \, \mathbb{E}_{x \sim \mathrm{Unif}(\mathrm{supp}(X))} \left[ \left| \hat{Y}(u) - \hat{Y}_x(u) \right| \right],$$

which measures the path-specific sensitivity of the model output with respect to the bias attribute $X$.

Similarly, in analogy to counterfactual accuracy, we introduce the *counterfactual $R^2$ score*:

$$R_{\mathrm{ctf}}^2 := 1 - \frac{\mathbb{E}_{u \sim U} \, \mathbb{E}_{y \sim \mathrm{Unif}(\mathrm{supp}(Y))} \left[ \left( y - \hat{Y}_y(u) \right)^2 \right]}{\mathbb{E}_y \left[ \left( y - \bar{Y}(u) \right)^2 \right]},$$

where $\bar{Y}$ denotes the mean of $Y$ over all counterfactually generated samples. This parallels the standard $R^2$ score, but is evaluated under counterfactual interventions on the $Y$ attribute.

*Table 6.* Counterfactual sensitivity $S_X$ and counterfactual $R^2$ scores across methods on dSprites. Lower $S_X$ indicates reduced spurious dependence on the bias attribute $X$, while higher $R^2_{\text{ctf}}$ indicates better counterfactual predictive performance.

| Method | $S_X$ | $R^2_{\text{ctf}}$ |
|---|---|---|
| sDISCO | 0.058 | 0.7181 |
| HSCIC | 0.064 | 0.6021 |
| CIRCE | 0.064 | 0.5939 |
| Adversarial | 0.084 | 0.4812 |

Since $Y$ has a standard deviation of 0.2 in the test set, we observe that all models manage to reduce dependence on the bias variable $X$, though to varying degrees. Recall from Tab. 2 in the main text that sDISCO ranked above HSCIC and CIRCE in terms of balanced accuracy. While this metric reflects a combination of debiasing and predictive performance, it does not disentangle which factor contributed more.

Counterfactual analysis provides this finer-grained perspective. Tab. 6 shows that sDISCO, HSCIC, and CIRCE all achieve comparable mitigation of bias effects, as indicated by similar $S_X$ values. However, their counterfactual $R^2$ scores reveal a different story: both HSCIC and CIRCE reduce dependence on $X$ at the cost of predictive accuracy on the task-relevant variable $Y$.

In summary, while balanced accuracy on a bias-balanced test set is useful for reporting overall performance, counterfactual path analysis enabled by SAM allows us to better understand *why* certain models fail. In this experiment, we see that HSCIC and CIRCE, although nearly matching sDISCO in debiasing effectiveness, sacrifice too much information from the true target variable $Y$.

### H.5. Runtime and Memory Analysis

To provide a comprehensive view of the computational overhead and scalability of the evaluated bias mitigation methods, we conducted a runtime and memory analysis on the FairFace dataset, with results summarized in Tab. 7. All benchmarks were executed on a single NVIDIA Titan RTX GPU (24 GB VRAM) paired with an AMD Ryzen Threadripper 1920X CPU. We employed 8 parallel data-loading workers. The input pipeline processes standard RGB images resized to $224 \times 224$ pixels, applying standard training augmentations (random horizontal flips and random affine transformations) prior to normalization. The reported mean epoch times encompass the full forward and backward passes using a ResNet-18 backbone in standard 32-bit floating-point precision.

*Table 7.* Mean epoch run time (in seconds) across varying batch sizes. Entries marked with *OOM* indicate that the method failed due to an Out-Of-Memory error on a Titan RTX (24 GB GPU). While the naive tensorized implementation of DISCO hits a memory bottleneck early, sDISCO successfully scales to large batch sizes, matching the speed of the most efficient baselines.

| Method | Batch Size | | | |
|---|---|---|---|---|
| | 64 | 256 | 768 | 1024 |
| GDRO | 21.46 | 20.94 | 25.03 | 29.38 |
| IRM | 22.78 | 20.93 | 25.25 | 28.78 |
| Fishr | 31.94 | 31.44 | OOM | OOM |
| c-MMD | 34.55 | 36.01 | OOM | OOM |
| CIRCE | 20.62 | 21.01 | 25.54 | 28.33 |
| HSCIC | 21.62 | 21.52 | 25.58 | 29.53 |
| $\text{DISCO}_m$ | 21.95 | 22.69 | OOM | OOM |
| Naive DISCO | 36.49 | 38.46 | 42.17 | OOM |
| sDISCO | 21.75 | 22.61 | 25.36 | 29.06 |

