# OpenReview forum: "DISCO: Mitigating Bias in Deep Learning with Conditional Distance Correlation"
_ICML.cc/2026/Conference — ICML 2026 spotlight_

### Official Review · Reviewer_bWXi · 2026-02-20

**Soundness:** 4
**Presentation:** 3
**Significance:** 3
**Originality:** 3
**Overall Recommendation:** 5
**Confidence:** 3

**Summary:**

This paper first introduces a general framework, the Standard Anti-Causal Model (SAM), under which one can study many (arguably all relevant) biased prediction problems. The authors use this framework to establish a conditional independence condition that holds when prediction biases vanish. They then introduce two estimators, $\textrm{DISCO}_m$ and $\textrm{sDISCO}$, for distance correlation, showing that these estimators are consistent. Both estimators can readily be used to reguralize standard deep-learning based prediction frameworks. The authors provide evidence for the utility of this regularization across a number of biased prediction tasks, showing competitive performance of their methods against competing approaches.

**Compliance With Llm Reviewing Policy:**

Affirmed.

**Final Justification:**

After the discussion phase, all of my raised concerns were fully addressed, reflected in my positive score. I recommend acceptance.

**Key Questions For Authors:**

1. The main question I have, which I also find to be the most critical for proper understanding of this work, is with regards to the practical implementation of your method. Looking at Eq. (23), one could naïvely assume that you have access to observations of $\mathbf{B}$ during training. However, if this is the case, I would directly know what a bias-inducing variable is, and conversely what can be kept as a causally stable predictor, no? Can you explain in detail what is the input to your training pipeline, what you have access to in terms of variables during training, and how the regularization terms act in detail, in practice? Does the model only see samples of images and targets and must learn an internal representation that "disentangles" the causally stable predictors from others?
2. Could you provide additional intuition (or references) on how the form that both of your proposed estimators take is motivated?
3. Why do you opt for the anti-causal prediction paradigm instead of the causal one? Are there any clear trade-offs between the two?
4. Is having an unbiased validation dataset representative of problems we expect to see in biased prediction? I am not enough of an expert to judge if this is standard, but to me it seems that there are also many situations where this is a strong assumption.

I believe the contributions the authors make in this paper merits publication, and generally lean towards this direction. I believe the questions I have raised are important to clarify, as I hope they can further strengthen this work. Given that they are adequately addressed, I will raise my score.

**Limitations:**

yes

**Strengths And Weaknesses:**

## Strengths
- The main technical contribution the authors make (introduction of consistent estimators of conditional distance correlation) overcomes a previously unresolved challenge.
- The unifying framework that is presented to study the problem at hand addresses two important points: 1) it provides the basis for motivating the premise of the paper and 2) it serves as the technical scaffolding to establish theoretical results. The framework allows to abstract away many (nuisance) details of specific bias cases and focus on their shared structure.
- The presentation of this paper is very nice, the main ideas are easy to follow and the sections flow nicely into each other.
- Explicit discussion of assumptions (Section 2.2) is very welcome.

## Weaknesses
- The discussion of the proposed estimators and where they are motivated from could be more verbose.
- Some key details of how the proposed regularizer functions in practice are missing for a complete understanding and deserve to be discussed in greater detail. Please see the Key Questions for details.

### Minor points
- Typo: line 422, left column, "out" should be "our"
- Proof sketches would be nice to have, even if the full proofs are in the appendix. If you need more space, I would recommend moving the details of each dataset in the experiments section to the appendix (and only keeping their respective graphs).
- Figure 3: the blue edge is very hard to visually distinguish from the other black edges. I would recommend to make it red.
- Figures should generally be at the top of a column, not in the middle.
- Can you explain the indexing difference between $ctf-DE$ and $ctf-IE$? Why are the subscripts $y_0, y_1$ swapped?

---

> ### Author Rebuttal · Authors · 2026-03-25
>
> We sincerely thank the reviewer for their constructive feedback, careful reading of our manuscript, and the collaborative tone of the review. Below, we address your questions and outline the corresponding updates we will make to the paper to improve clarity.
>
> **Q1**
> Yes, we assume access to the bias variable $B$ during training. Our work operates within the well-established "informed bias mitigation" paradigm, which is standard in fairness auditing and fields like medical imaging (where metadata like scanner type or patient sex is typically documented and must be accounted for without discarding data). This assumption is made by all methods within this field, including all of our (high-impact) baselines.
>
> In practice, our pipeline works as follows: The model only takes the input $X$ to generate the prediction $\hat{Y}$. It does *not* see $B$ to make its prediction, meaning it must learn an internal representation that isolates the causally stable predictors. However, during the backward pass while training, our regularization term calculates the conditional distance correlation between the model's predictions $\hat{Y}$ and the known bias labels $B$, conditioned on $Y$. This penalizes the network if its predictions leak information about the bias. We will add a dedicated paragraph in the methodology section detailing this exact forward/backward pipeline to prevent any reader confusion.
>
> **Q2**
> Classical conditional distance correlation is a powerful statistical tool, but its standard formulations are computationally prohibitive and incompatible with neural network training. The core intuition behind DISCO$_m$ and sDISCO was to adapt this measure into a differentiable, mini-batch compatible format. By computing centered pairwise distances between samples in a batch, we can efficiently estimate how much the model's predictions rely on the bias, conditioned on the target, allowing us to penalize that dependence directly via backprop. We will expand the intro of Section 3 to include this high-level intuition.
>
> **Q3**
> We opted for the anti-causal paradigm because it most accurately reflects the data-generating process for the vast majority of computer vision and NLP tasks, see Schölkopf et al., 2012. For instance, in medical imaging, a classical textbook inverse problem, the underlying disease (the target $Y$) physically causes the manifestations in the pixels (the input $X$). Similarly, an object's underlying concept causes its visual features. Modeling this as anti-causal allows us to properly map how spurious mediators or confounders leak into the observed data. We will add a brief discussion to Section 2.1 to clarify this trade-off and justify the assumption.
>
> **Q4**
> You are completely right that this is a strong assumption. However, utilizing either an unbiased validation set or having bias annotations to compute Worst Group Accuracy (WGA) is the accepted standard protocol in the bias mitigation literature. Even in "unsupervised" bias mitigation (where $B$ is not seen during training, such as "Just Train Twice", Liu et al.), methods typically require bias labels for validation and model selection.
>
> Because our framework uniquely handles both classification and regression tasks (where WGA cannot be computed), we initially relied on bias-balanced OOD validation sets to keep our model selection logic consistent. However, for the MNLI dataset, creating a balanced validation set would have required discarding an unacceptable amount of data. Therefore, we followed the standard protocol of using bias labels to compute WGA for that specific task.
>
> ### Minor Points
>
> **Indexing Difference ($ctf-DE$ vs $ctf-IE$)**
> Great catch! The difference is intentional. We adopted it directly from the established notation by Plecko & Bareinboim (2022) to remain consistent with causal fairness literature. Conceptually, $ctf-DE$ measures the effect of changing the target $y_0 \to y_1$ while holding the mediator constant at its natural value under $y_0$. Conversely, $ctf-IE$ holds the target constant at $y_1$ and measures the effect of changing the mediator from its value under $y_1$ to its value under $y_0$.
>
> **Formatting and Structural Updates**
> *   **Typo:** Fixed "out" to "our" on line 422.
> *   **Figure 3:** We have changed the blue edge to red for better visual distinction.
> *   **Figure Placement:** We will adjust the formatting to ensure figures appear at the top of the columns.
> *   **Proof Sketches:** We agree that having proof sketches in the main text improves readability. We will move the detailed dataset descriptions from Section 4.1 to the Appendix to make space for the proof sketches in the main body.
> *   **Moving Dataset Details:** We move the notation heavy SCMs of our datasets into the appendix. This will create enough room for changes proposed by you and the other reviewers. Thanks for this suggestion.
>
> We hope these clarifications address your concerns, and we deeply appreciate your help in strengthening our work.

---

> > ### Author Rebuttal · Reviewer_bWXi · 2026-04-01
> >
> > All points have been adequately addressed. Thank you.

---

> > > ### Author Response · Authors · 2026-04-03
> > >
> > > We thank the reviewer for their highly constructive and positive feedback. Their comments helped us in fixing presentation issues while also improving other important aspects of the paper.

---

### Official Review · Reviewer_jUq1 · 2026-03-12

**Soundness:** 3
**Presentation:** 3
**Significance:** 3
**Originality:** 2
**Overall Recommendation:** 4
**Confidence:** 4

**Summary:**

This paper studies bias mitigation in supervised learning from a causal perspective. The authors assume an anti-causal setting and introduce a structural model that separates variables into target-related components and bias-related components (including backdoor variables and mediators). They define causal stability as relying on stable task signal rather than bias pathways, and use the observable criterion Ŷ(predicted label) ⟂ B(bias) | Y(true label) as the central goal. Experiments on six datasets across vision and NLP compare against robustness, fairness, and shortcut-mitigation baselines and report improved worst-group/bias-robust performance with competitive average accuracy.

**Compliance With Llm Reviewing Policy:**

Affirmed.

**Final Justification:**

The rebuttal addresses part of my concerns clearly, particularly by clarifying the causal motivation and practical implementation of the proposed conditional distance-correlation regularization. The work remains well-structured and empirically credible, with solid evaluation across diverse datasets, and its causal framing provides a coherent perspective on bias mitigation.

However, my overall assessment is unchanged because the core contribution is still largely incremental relative to prior fairness and conditional-independence-based approaches, and the practical applicability remains limited to settings where bias variables are explicitly observed and predefined. For example, extending this approach to large-scale industrial recommendation systems with large feature space would be challenging. I am maintaining my rating as weak accept.

**Key Questions For Authors:**

- Positioning versus prior separation-based robustness work
 How does the paper’s central criterion, Y^⊥B∣Y, differ in substance from the separation principle studied in prior work such as Makar and D’Amour? Since the observable criterion appears very similar, it would help if the authors clarified what the main difference is in the current work?

-  Role of the page-3 theory in the final method
 Could the authors clarify which parts of the effect decomposition in Section 2 are essential for deriving the final method, and which parts are primarily interpretive? In particular, it is not fully clear how central Proposition 2.1 and Theorem 2.4 are to the operational objective, versus serving mainly as motivation for the conditional independence constraint.

- Scope and applicability of the method
 The title and framing are broad, but the method seems to apply specifically to settings where the relevant bias variables are observed and specified in advance. Could the authors clarify this scope more explicitly, and discuss how they see the method fitting into the broader landscape of bias mitigation methods for deep learning?

**Limitations:**

- The practical scope is much narrower than the framing suggests. The method assumes that the relevant bias variables are already known, observed, and labeled. That is a strong requirement, because in many real problems the hardest part is not removing bias once identified, but identifying what the nuisance variable actually is. The framework also relies on an anti-causal setup and sufficient overlap between target and bias variables, which further limits where the method can be applied. So the paper is useful in a specific observed-bias regime, but not a general solution to bias mitigation.

**Strengths And Weaknesses:**

Strengths

- The paper presents a well-structured and conceptually unified formulation of bias mitigation. A major strength of the work is that it does not present debiasing as just another regularizer or empirical trick, but instead starts from an explicit causal formulation of the prediction problem.

- The  proposed conditional distance-correlation regularization-is practical and well motivated.  The paper does more than propose a causal perspective; it introduces an explicit training mechanism that can be implemented in gradient-based learning. The estimator construction is designed to address a critical gap, namely how to enforce the target conditional-independence criterion in settings with mixed variable types and modern model classes.

- The empirical evaluation is broad and adds credibility to the method. The paper evaluates the approach across a fairly diverse collection of datasets and bias settings, which helps demonstrate that the method is not tied to a single benchmark or narrow application.

Weaknesses

- The novelty of the proposed approach  is incremental. The conditional independence criterion Ŷ ⊥ B | Y is not conceptually new and prior work in fairness and robustness has employed similar separation constraints. The central conditional-independence idea is closely related to prior work, and the conditional distance-correlation foundation also appears to build on existing theory. Overall;  the paper’s main contribution is less on the introduction of a new principle and more on the provision of a causal interpretation.

- The practical scope is narrower than the title and framing imply. The method applies in a setting where the relevant bias variables are observed, specified in advance, and suitable for explicit regularization. That is an important setting, but it is much narrower than “mitigating bias in deep learning” in the broad sense suggested by the title. In many realistic problems, identifying the relevant nuisance variable is itself challenging, and this framework largely assumes that problem is already solved. The paper acknowledges this setup, but the overall framing still makes the contribution sound more general than it really is.

---

> ### Author Rebuttal · Authors · 2026-03-25
>
> We sincerely thank Reviewer jUq1 for the constructive review and the "Weak Accept" recommendation.
> Your feedback regarding our framing and positioning is highly valuable. We address your specific questions below and hope to clarify the dual nature of our contribution.
>
> ### 1. Positioning Versus Prior Separation-Based Work
>
> We dedicate **Appendix B.2** specifically to contextualizing our work against Makar & D'Amour, Veitch, and Puli. To summarize the substantive difference:
> *   **Makar & D'Amour (2022)** operate primarily at the distributional level (Pearl’s Rung 1: Association), describing *what* statistical property a robust model should exhibit.
> *   **Veitch et al. (2021)** operate at the counterfactual level but require unit-level counterfactual invariance ($f(X(z)) = f(X(z'))$), which relies on unobserved potential outcomes and is difficult to enforce practically.
> *   **Our Contribution:** SAM operates at the structural and counterfactual levels (Rungs 2 and 3) but bridges the gap to practice. By explicitly mapping the information flow, we provide the underlying *causal mechanism* proving exactly *why* the separation principle works. We show mathematically that blocking specific spurious paths in the graph necessitates the resulting distributional independence, making an abstract ideal practically enforceable.
>
> ### 2. Role of the Page-3 Theory
>
> Proposition 2.1 and Theorem 2.4 are foundational to the method, not just motivational.
> *   **Proposition 2.1** provides the mathematical decomposition of total variation into direct, indirect, and spurious counterfactual effects.
> *   **Theorem 2.4** leverages this to prove that enforcing $\hat{Y} \perp B \mid Y$ via Maximum Likelihood Estimation mathematically forces the indirect and spurious effects to zero, thereby maximizing the stable direct effect.
> These proofs are essential because they guarantee that our optimization objective (Eq. 7) is not merely an empirical heuristic, but a mathematically sound mechanism for achieving causal stability under SAM.
>
> ### 3. Scope and Applicability
>
> We completely agree with your assessment. Identifying unobserved nuisance variables is a distinct and massive challenge in AI. Our method specifically targets the "Discriminative Causal Representation Learning with observed bias" regime (as detailed in Appendix B.1). However, we argue this is a highly critical and broad setting: encompassing fairness auditing (where sensitive attributes are legally required to be known), medical imaging (where scanner types/hospital metadata are recorded), and standard safety benchmarks.
>
> **Action taken:** We appreciate this constructive push. To ensure we do not over-claim, we will gladly update our title to *“DISCO: Mitigating Observed Bias in Deep Learning...”* and explicitly clarify this boundary condition in the abstract. We might also add this point to the limitations section (which we will introduce). But we want to highlight that observed bias mitigation is a deeply established field, and all the limitations also apply to all these methods, including all of our (high-impact) baselines.
>
> ### 4. Originality and the Algorithmic Contribution
>
> We respectfully ask you to reconsider the Originality score in light of a crucial algorithmic distinction that may not have been fully emphasized in an initial read.
>
> While the principle of conditional distance correlation was mathematically defined in 2015, **it was computationally prohibitive and fundamentally unusable for modern, gradient-based deep learning (DL).** Prior to our work, enforcing this constraint over continuous, mixed-type, and multivariate data inside a neural network optimization loop was an unsolved systems and algorithmic challenge. We also added runtime and memory analysis into appendix of camera-ready draft, see some numbers in our response to reviewer pbVU (see point W7).
>
> Our primary originality lies in bridging this exact gap. The introduction of the DISCO$_m$ and single-shot sDISCO estimators is a major algorithmic step forward. We successfully designed estimators that:
> 1.  Are fully differentiable and compatible with backpropagation.
> 2.  Scale seamlessly to multi-bias and mixed-type scenarios (unlike standard DRO or IRM).
> 3.  Require drastically fewer hyperparams and computational overhead compared to SOTA RKHS methods like HSCIC and CIRCE.
>
> The originality of this paper is dual: we provide a rigorous causal theoretical foundation derivation, **and** we invent the scalable algorithmic tools (DISCO) to actually compute it in DL architectures.
>
> ### Conclusion
> We hope our commitment to refining the title/scope, combined with the clarification of our algorithmic breakthroughs regarding distance correlation, addresses your core reservations. If you feel we have successfully clarified the originality of our estimators and the specific theoretical mechanisms we contribute, we kindly ask if you would consider updating your Originality score and overall recommendation.

---

> > ### Author Rebuttal · Reviewer_jUq1 · 2026-04-03
> >
> > Thank you for the detailed response. I will maintain my rating of 'weak accept' due to the narrow problem scope and the incremental nature of the contribution.

---

### Official Review · Reviewer_Uimw · 2026-03-17

**Soundness:** 3
**Presentation:** 2
**Significance:** 3
**Originality:** 3
**Overall Recommendation:** 5
**Confidence:** 4

**Summary:**

This paper studies bias mitigation in deep learning from a causal perspective. The authors argue that models often learn the “easy shortcut” instead of the real task signal. To address this, It introduces a causal framework called  the Standard Anti-Causal Model (SAM). In this setup, the label $Y$ is viewed as generating the input $X$, and the authors separate bias variables $B$ into backdoor-type variables $Z$ and mediator-type variables $W$. Then **they derive a simple target condition for a good model: the predictor $\hat{Y}$  to be conditionally independent of observed bias attributes $B$ given the label $Y$, written as $\hat{Y} \perp B \mid Y$.** The paper argues that if this holds, then indirect and spurious effects vanish and the predictor becomes “causally stable.”

The main technical contribution is that **they turn this idea into a training regularizer:**  The difficulty is that checking conditional independence directly is hard, especially inside deep learning. So they propose two differentiable estimators of conditional distance correlation, called DISCOm  and its cheaper variant sDISCO that can be used as regularizers in standard gradient-based training, turning the constrained objective into prediction loss plus a dependence penalty between predictions and bias conditioned on the target.
These are designed to measure how much the model output $\hat{Y}$ still depends on the bias variable B after conditioning on $Y$. If that dependence is near zero, the model is closer to the desired criterion.


Empirically, the paper evaluates the approach on six vision and NLP benchmarks and reports performance that is generally competitive with or better than prior debiasing baselines.

**Compliance With Llm Reviewing Policy:**

Affirmed.

**Final Justification:**

My concern has been somewhat addressed, so I will raise my score to accept.

**Key Questions For Authors:**

See the Weaknesses

**Limitations:**

yes

**Strengths And Weaknesses:**

# Strengths
- The paper tackles an important problem—shortcut learning and bias-sensitive prediction—which is central to robustness, fairness, and out-of-distribution generalization. Its main significance is that it offers a unified causal perspective on different types of bias and connects that perspective to a practical training objective usable in standard deep learning pipelines. That makes it relevant both conceptually and methodologically.
- The proposal of DISCOm and sDISCO is a meaningful technical contribution.
- The paper is clearly written and well structured overall. The narrative is easy to follow.


# Weaknesses

- Result is conditional on a specific causal world. The theorem itself says the implication holds under SAM and no unobserved backdoor paths between input $X$ and target $Y$.

- In the main paper, the estimators are presented as if their consistency is proved, but the appendix material is labeled as a “Proof Sketch” for DISCOm and an “intuitive proof sketch” for sDISCO (Proposition 3.6). That is a meaningful rigor gap for theorem-level claims and  it weakens confidence in the completeness of the theoretical development.

- Section 2.2 refers to “Theorem 1”.  the appendix related-work section also refers to “Theorem 1”. But there is no Theorem 1.

- The paper says DISCOm and sDISCO “consistently outperform” or are “superior or competitive,” and the appendix even says the proposed predictors “consistently lead the performance tables or, at worst, rank second.” That is too strong as written. By their own reported numbers, CIRCE is best on Waterbirds, GDRO slightly beats DISCOm on MNLI, and sDISCO is not always first or second across tables. The results are good and competitive, but the summary language should be toned down.

---

> ### Author Rebuttal · Authors · 2026-03-25
>
> We thank Reviewer Uimw for the constructive feedback and the "Weak Accept" recommendation. We are glad you found the problem important and recognized DISCOm and sDISCO as meaningful technical contributions.
>
> Below, we address your concerns. We are committed to making these changes to strengthen the camera-ready version of the paper.
>
> **1. "Result is conditional on a specific causal world (SAM)"**
> We completely agree. This is a fundamental limitation of our method and of observational bias mitigation methods in general. We explicitly acknowledge this in Section 2.2 ("Unobserved Confounding"), noting that if unobserved paths exist, the predictor remains biased, but blocking all *observed* paths remains the best achievable approximation of causal stability in general. To ensure this limitation is not missed, we will elevate this point by adding a dedicated sentence to both the Introduction and a separate limitations section, explicitly stating that our theoretical guarantees depend on the completeness of the observed bias set $B$. Yet we again want to highlight that we made our assumptions (thereby our limitations) very explicit in section 2.2. However, we agree to make it stronger by creating a new, additional limitations section.
>
> **2. "Proof Sketch" terminology vs. theorem-level claims in the main text.**
> We thank the reviewer for pointing out this discrepancy in our terminology. We completely agree that labeling the appendix sections as "sketches" creates a mismatch with the phrase "we prove" in the main text.
>
> To clarify, the consistency of DISCOm and sDISCO relies directly on the well-established asymptotic properties of classical kernel regression (specifically, the Nadaraya-Watson estimator), U-statistics, and Slutsky's theorem. Because these foundational proofs are standard in the literature, we opted to provide structural sketches that map exactly how these established theorems apply to our novel conditional distance correlation formulation, rather than re-deriving classical kernel convergence proofs from first principles.
>
> We recognize that this choice of presentation must be better signaled in the main text to maintain academic rigor. In the camera-ready version, we will retain the step-by-step mathematical logic in the appendix, but we will revise the main text language. Specifically, we will replace "We prove that..." with "Building on standard kernel regression assumptions, we demonstrate the consistency of..." and remove the "theorem-level" framing of Propositions 3.5 and 3.6 to perfectly align the reader's expectations with the appendix material. But we emphasize again, that the formal proofs are unnecessarily complex and require building up either heavy analytical machinery (epsilon-delta) or heavy measure theoretic convergence boilerplate, without adding anything actually meaningful. All the details are thoroughly handled in wide-spread statistics and measure theory literature, especially in the field of non-parametric estimation. Apart from that, we will further provide a more direct proof for the sDISCO variant which we think can be expressed more cleanly and directly.
>
> **3. Reference to "Theorem 1".**
> We thank the reviewer for catching this oversight. This is an artifact from an earlier draft of the paper. All references to "Theorem 1" in Section 2.2 and the related-work appendix were intended to reference "Theorem 2.3" (Conditional Independence $\Rightarrow$ Causal Stability). We will fix this typo immediately.
>
> **4. Overstated performance claims ("consistently outperform", "superior").**
> We appreciate the reviewer keeping our claims grounded. You are entirely correct that our language is quite strong given the nuances of the results, specifically regarding CIRCE's performance on Waterbirds and GDRO's slight edge on MNLI. We directly adopted the language from comparable papers in the field of computer vision and NLP bias mitigation. Yet, reading your analysis on that, we agree to tone down the summary language in the Abstract, Section 4.3, and the Appendix.
>
> Specifically, we will change "consistently outperform or are superior" to more accurate phrasing, such as: *"Our methods perform competitively with SOTA baselines, frequently achieving top or second-best performance across diverse vision and NLP tasks, while requiring fewer hyperparameter choices."*
>
> We hope these revisions address your concerns and successfully close the rigor gap you identified. We are grateful for the close reading of our theoretical claims and results (which is often overlooked).
>
> If you are convinced that our updates will improve the presentation (you gave a 2), we would be very happy with an according update of the score, and potentially the overall recommendation. Please let us know, if we addressed your concerns well enough with our propositions made here, and we appreciate concrete follow-ups if we missed any of your points you made above.

---

> > ### Author Rebuttal · Reviewer_Uimw · 2026-04-01
> >
> > My concern has been somewhat addressed, so I will raise my score to accept.

---

> > > ### Author Response · Authors · 2026-04-03
> > >
> > > We thank the reviewer for their highly constructive approach to our paper, and acknowledging the resolution of their key issues. We thereby improved the proof presentations, especially finding a much clearer presentation of the sDISCO proofs, and adjusted claims and language at important places. We also appreciate them catching details such as phantom references that we now fixed.

---

### Official Review · Reviewer_pbVU · 2026-03-20

**Soundness:** 3
**Presentation:** 3
**Significance:** 3
**Originality:** 3
**Overall Recommendation:** 5
**Confidence:** 4

**Summary:**

In this work, the authors study how to train deep learning that are unbiased towards certain known features through a causality lens. In particular, the authors assume an anti-causal setup where the model predicts a target variable using features _caused_ by it (hence anticausal). To this end, the authors propose a Standard Anti-Causal Model (SAM) that encapsulates this setting, and study under which conditions the predictions will be robust to changes on the undesired attributes (biases). Then, the authors propose two ways of efficiently induce this behaviour in a model and compare their method with respect to other models in a number of semi-synthetic experiments.

**Compliance With Llm Reviewing Policy:**

Affirmed.

**Final Justification:**

During my review, I had some concerns regarding the soundness of some of the theoretical claims as well as empirical results. After a really poor initial rebuttal by the authors, and a assertive reply on my behalf, I am happy to see that the authors' follow-up were in way better-terms, try to understand me rather than challenge me, and that my concerns have been properly addressed in this second reply.

I still keep my concerns regarding the overly complicated presentation of the work, which could be significantly simplified and there is no good justification to not do so, but I will let it slip through as it does not undermine the paper contributions.

I have changed my score to Accept to reflect the above.

**Key Questions For Authors:**

I am happy if the authors can respond to the points raised above.

Additionally, I **strongly** encourage the authors to add some text in the _main_ text discussing potential ethical concerns, especially given that they decided to use one experiment to predict the _sex_ of a person based on a _picture_ where _skin tone_ is the bias term (deduced from the reported race!)

**Limitations:**

As far as I can see, there is no explicit limitation section, which would be a nice addition to have. For example, I am not sure if it is somewhere explicitly mentioned that ones needs to know in advance what features are M, Z, and X.

**Strengths And Weaknesses:**

**Strengths:**
- S1. The problem tackled on this work is important and of interest to the broad machine learning community.
- S2. The final proposed method is sensible, the estimators well-detailed and the empirical results look quite positive.
- S3. There seem to be enough details to properly reproduce all results.

**Weaknesses:**
While I like this submission, I think there are some considerable problems in different departments that, of fixing them, would significantly strengthen this manuscript. Namely:
- W1. Writing. The writing could be improved, as there are some unnatural jumps that difficult reading. For example, from the beginning there are mentions to "causal stability" as a well-known concept, but it is defined in page 3. Other examples are that the justification of section 2 (lines 119-122) cannot be "we want to reach the result we spoiled in the intro", or that there is a section dedicated to analyze assumptions (2.2) that, unless I missed them, are not introduced before that section.
- W2. Unnecessary complexity. This is quite a big concern I have. I feel the presentation is overly complicated. For example, I do not understand the need to introduce the SAM model and distinguish between Z and  M, if then I am going to group all as B (and later divide also M in two?). Everything could be simply grouped as (X, Z, Y) ---at the end, we are the ones choosing what is what---and say that we want a transformation of X (the prediction, or $\tilde X$ in general) that is independent of Z given Y. To my understanding, the entire paper revolves around the idea of disentangling X from Z given Y, but it is strangely obscured.
- W3. Causal claims. I find some more consideration should've been put in some causality claims. For example, the general model in Figure C.1 includes a collider variable that is not mentioned in the main paper, and the simplified SAM model could include Z as a collider (since the edges are undirected) and therefore open backdoor paths upon conditioning (rather than blocking them). Or calling the "direct" causal effect between target and prediction, when clearly it is not direct given the assumed graph. Another example are the claims regarding hidden confounders ("unknown biases cannot be removed, but blocking all known biases is the best we can and should do"), which is directly false as extensively proven by the works of Miao and Blei among others in proximal identifiability.
- W4. Given the above, I am also a bit concern regarding the novelty of the method as of what it actually achieves. I have the hunch (and I'd love to discard) that replacing the DISCO estimator in Eq. 23 by a conditional covariance estimator would obtain a similar effect.
- W5. Regarding experiments, I find it a bit discouraging that there are not truly real experiments, where biases do not have to be introduced synthetically. It really goes against the idea that the proposed method can be applied to solve a real problem that practitioners face in the wild.

Minor:
- W6. There is a disproportionate jump between sections 2 and 3. First assumptions are made to make things simple (e.g. only discrete variables) and in the next section heavy math is thrown to the reader without anticipation.
- W7. It would be really beneficial to report the training and inference times of all methods, to see why we need these fancy estimators in the first place (especially since the initial argument is that distance correlation was expensive to use) and that a cheaper estimator was proposed after the first one.

---

> ### Author Rebuttal · Authors · 2026-03-25
>
> We thank the reviewer for their thoughtful, respectful, and constructive feedback, particularly for acknowledging the positives of our work.
>
> **Ethical Concerns regarding FairFace and Skin Tone**
> We completely align with the reviewer that ethical considerations are essential, and we share the exact concerns regarding the use of "race" as an attribute. This is precisely why we explicitly stated in Appendix F.1.4 that we "do not endorse the conceptualization of race as used in FairFace: >race< is ontologically unstable, lacks scientific grounding, and reproduces historically problematic constructs". Because the original dataset creators (who are computer scientists, not social scientists) used unconventional categorizations, we mapped their label to the physical proxy of "skin tone". We did this not to keep the dataset's original structure, but to demonstrate how our method can effectively *remove* this highly problematic bias. We will gladly move this ethical discussion to the main text.
>
> **W2**
> The distinction between mediator variables (W) and backdoor variables (Z) is not an unnecessary complication; these are variables established in causal literature (e.g., Pearl, 2009; Shpitser & VanderWeele, 2011): mediators and confounders/colliders carry profoundly different semantic and graphical implications. SAM puts these into an anti-causal prediction context.
>
> The distinction is also crucial for the bias correction with DISCO. Backdoor variables (Z) **must** be included in B. Whereas mediator variables (W) **can** be included in B. The decision for their inclusion is application and domain dependent. We kindly refer the reviewer to section 2.2 "selective mediation" for a detailed discussion.
>
> **W3**
> * **Colliders:** In our graph (Figure C.1), Z explicitly represents variables on **open** **backdoor** paths. The reviewer might have missed that the collider is outside of the box/set of Z, and the collider is already conditioned on, thus the path is already open. Z contains only nodes that are already on open paths. We will add details to the main body to clarify this more explicitly.
> * **Direct Effects:** We use the term "counterfactual direct effect" (ctf-DE) exactly as defined in the causal fairness literature. It refers to the direct effect **relative** to the specified pathways in the assumed graph. We made our convention explicit.
> * **Hidden Confounders:** The reviewer makes an interesting point regarding proximal identifiability (e.g., Miao and Blei). However, proximal causal inference requires observing valid *proxies* for the unobserved confounder. In our strict observational setup, where no such proxies are assumed to exist, blocking known biases remains the theoretical limit. We will revise our claim to explicitly acknowledge proximal methods as a complementary approach when proxy variables are available.
>
> **W4**
> Replacing DISCO with a standard conditional covariance estimator would fundamentally fail in this setting. Standard conditional covariance only measures *linear* relationships and is one dimensional. Deep neural networks learn highly non-linear representations. Distance correlation (and our DISCO) uniquely captures non-linear dependencies across arbitrary dimensions. This capacity to penalize complex, non-linear shortcut correlations is exactly why DISCO is required over traditional covariance.
>
> **W5**
> We respectfully clarify that while dSprites and Blob are synthetic (which is strictly necessary to compute the exact ground-truth counterfactuals for our pathway analysis in Section 4.4), Waterbirds, FairFace, YaleB, and MNLI are widely accepted, standard real-world benchmarks. Waterbirds is the de facto standard in bias mitigation, FairFace uses real human faces, and MNLI uses real natural language. The biases in these datasets (backgrounds, demographics, negation words) are real, naturally occurring spurious correlations.
>
> **W1 & W6**
> We appreciate this feedback. We will add a brief, intuitive definition of "causal stability" in the introduction and include a transitional paragraph in Section 3 to better bridge the jump. We further will create a better introduction for section 2 itself, referencing our assumptions subsection and contextualizing it better.
>
> **W7 & Limitations**
> Our methods are faster and more mem. efficient than naive implementation: $23.35$s$\pm 0.35$ ($DISCO_m$) vs $44.08$s$\pm 0.49$s (naive, Wang et al.) vs $33.31$s$\pm 0.32$ (Fishr) mean time per train epoch. On larger batches, naive does not run at all (memory errors on 24GB VRAM). We will add all runtime and memory comparisons, incl. hardware details to appendix (no space here). Regarding limitations, Section 2.2 currently discusses assumptions (limitations). We will add a detailed "Limitations" section/paragraph for better visibility.

---

> > ### Author Rebuttal · Reviewer_pbVU · 2026-04-02
> >
> > I appreciate the author's response, which have clarified some of my concerns. I especially appreciate the clarifications regarding time and memory complexity, as well as the use of FairFace. I had a strong believe the authors did not mean it in bad faith, otherwise I would reported it.
> >
> > However, my concerns on the theoretical and experimental parts have not been resolved. I'd appreciate it if the authors assume reviewers are knowledgeable of the topics they review and try to understand their concerns, rather than re-explain them what it is already written in the manuscript.
> >
> > **W2** I know the difference between mediator and backdoor variables, but I still fail to see how this distinction is crucial at all wrt. DISCO. The proposed algorithm does not treat them differently unless I missed something, and narrative could've been simplified a lot by simply assuming practitioners have already separated between variables to unbias from (B) and the rest (X). (Of course, assuming the causal links Y <-> B <-> X and Y -> X)
> >
> > **W3**
> > - **Colliders** I am capable of seeing that "coll" it's outside the box right below it. My concern was regarding its disappearance in the simplified model. Is "coll" assumed to be part of "X" in the simplified version? Or is "coll" assumed to be equal for every sample so that it does not need to be taken into account during training?
> > - **Direct Effects** I understood that the authors meant cf direct effect math-wise, yet there is not a single instance in the text where "counterfactual" appears before "direct effect" besides the title of the paragraph in line 142.
> > - **Hidden Confounders** I am fully aware of what is a proxy. In the general worst case, you are right, but nothing as far as I can see avoids the set X to contain proxy variables that could be leveraged.
> >
> > **W4** I do agree that DISCO can capture non-linear relationships that covariances cannot. I never said otherwise. What I am unsure is how much of a difference capturing these non-linear relationships would empirically make over adding the correlation between the Y-conditioned correlation between B and Y hat as a penalty term. Could the authors run that experiment to prove me wrong? (with the same budget for the parameter $\lambda$)
> >
> > **W5** I have worked with those real datasets in the past, but maybe I misunderstood it. Is it the case that the equations under YaleB, FairFace, and Waterbirds were inferred from the actual unaltered data? Are those equations describing how the datasets were re-sampled? Or are those real datasets which were complemented with synthetic colliders variables?

---

> > > ### Author Response · Authors · 2026-04-03
> > >
> > > **Updated:** direct effect and proxy, see below
> > >
> > > We thank the reviewer for their follow-ups and challenging us to further improve our argumentation and our ablation results.
> > >
> > > **W2** We agree that for an expert practitioner who understands that different variable types can be treated the same way upfront, this narrative could indeed be simplified. But we wanted to arrive at this conclusion by a formal treatment in the first place. We wanted to formally show that given the assumptions and setups of this paper, we can treat them equally. From this point on, we exactly continue as reviewer suggested. For many readers and also for ourselves in the beginning, this result was not obvious. Some papers dedicated their time to mitigate confounders in deep learning settings (Neto, 2020; Zhao et al., 2020), and others treated collider biases explicitly (Darlow et al., 2020). We consolidate these views and show that these seemingly different cases can be treated equally, also extending on unwanted mediation effects. And we arrive at the simple view point via a formal argumentation.
> > >
> > > **W3**
> > > - colliders: Yes, essentially the colliders are latently conditioned on due to hidden sampling bias effects (meaning they are effectively fixed/equal for the observed dataset). We dropped the already conditioned collider that is hidden and out of control of the practitioner. The only important effect it has is leaving us with an additional, unwanted open path. We only observe the variables in $\mathbf{Z}$, that lie on these, and other, open paths that we now must close. We can clarify this better in both figures and also extend on this in the main body, since we might have more space for a camera-ready version.
> > > - direct effects: thanks, we now see what the reviewer means. We propose to entirely rename the ctf-DE to ctf-stable or similar. We will thereby remove any kind of ambiguity and definitions that deviate from the norm. If the reviewer has a better recommendation, we are happy to hear it.
> > > - Proxies: The reviewer makes a fair point that X could theoretically contain proxy variables. Our framework, however, strictly assumes that X does not contain valid proxies for unobserved confounders. We will explicitly add this assumption regarding X to Section 2.2 to prevent any ambiguity, while also discussing proxies in more detail in the exact same section. We can additionally also adjust the language that removing the known biases is the best we can do given there are no proxies; making this explicit again will make it more precise - we definitely agree on that.
> > >
> > > **W4** Thanks for the clarification. We performed the requested ablation replacing DISCO with a correlation penalty for dSprites and FairFace, so we have both a regression and classification task. Note that conditional correlation is non-trivial to implement, we thus include two versions: partial correlation and stratified correlation (quantile binning over Y for regression). We gave the correlation loss version a larger space for $\lambda$: instead of 6 (see DISCO) we gave them 16 each. The stratified correlation also got in total 3 bin hyperparameters [2,3,4] (for the regression case. The classification task does not need this as it conditions on discrete values). We will add all the details and results of this ablation to the appendix. The results are:
> > >
> > > | Method | dSprites | FairFace |
> > > | :--- | :---: | :---: |
> > > | Stratified | $0.58 \\pm 0.016$ | $0.835 \\pm 0.004$ |
> > > | Partial | $0.51 \\pm 0.014$ | $0.830 \\pm 0.006$ |
> > >
> > > As shown in the table above, replacing DISCO with standard conditional correlation estimators leads to a performance drop. For context, our DISCO$_m$​ achieves 0.688 on dSprites and 0.860 on FairFace, outperforming both the Stratified and Partial correlation baselines. This confirms empirically that capturing non-linear dependencies is crucial. Especially our DISCO variants are out of the box applicable to multi-variable settings while the correlation based ones need again special treatment, particularly if the variables are of different types (highly non-trivial). Last, as a possible future work, we also want to highlight that DISCO, CIRCE, HSCIC can be applied to unbias high-dimensional feature embeddings with multi-dimensional conditions, while conditional correlation is limited to one dimensional triplets. We will add this to the conclusion.
> > >
> > > **W5** For Waterbirds and MNLI, the equations and graphs are inferred from the actual unaltered data. For YaleB and FairFace, the equations describe how the real datasets were re-sampled to introduce selection bias (colliders). We did not complement them with synthetic variables, but rather sub-sampled the real data to make the tasks harder and more biased. As the reviewer knows, this is standard practice also for other bias datasets (CheXpert/MIMIC-CXR: manually sub-sampling X-rays to create correlations with scanner, Camelyon-17: manually sub-sampling cancer images to correlate with hospital, etc.)

---

### Decision · Program_Chairs · 2026-04-30

**Decision:**

Accept (spotlight)

**Comment:**

The authors tackle bias mitigation deep learning from the perspective of casual inference using graphical models. Their proposed conditional independence criterion is shown to mitigate several types of dataset bias in vision and natural language tasks. Reviewers were uniformly positive about the paper, praising the principled estimation approach and practical implementation. I recommend accept.